# Tooth enamel nitrogen isotope composition records trophic position: a tool for reconstructing food webs

Jennifer N. Leichliter [1,2,3,10 ✉], Tina Lüdecke [1,2,4,10 ✉], Alan D. Foreman [1], Nicolas Bourgon [5], Nicolas N. Duprey[1], Hubert Vonhof [6], Viengkeo Souksavatdy[7], Anne-Marie Bacon[8], Daniel M. Sigman [9], Thomas Tütken [3] & Alfredo Martínez-García [1]

Nitrogen isotopes are widely used to study the trophic position of animals in modern food webs; however, their application in the fossil record is severely limited by degradation of organic material during fossilization. In this study, we show that the nitrogen isotope composition of organic matter preserved in mammalian tooth enamel ($\delta^{15}N_{enamel}$) records diet and trophic position. The $\delta^{15}N_{enamel}$ of modern African mammals shows a 3.7‰ increase between herbivores and carnivores as expected from trophic enrichment, and there is a strong positive correlation between $\delta^{15}N_{enamel}$ and $\delta^{15}N_{bone-collagen}$ values from the same individuals. Additionally, $\delta^{15}N_{enamel}$ values of Late Pleistocene fossil teeth preserve diet and trophic level information, despite complete diagenetic loss of collagen in the same specimens. We demonstrate that $\delta^{15}N_{enamel}$ represents a powerful geochemical proxy for diet that is applicable to fossils and can help delineate major dietary transitions in ancient vertebrate lineages.

[1] Organic Isotope Geochemistry Group, Climate Geochemistry Department, Max Planck Institute for Chemistry, 55128 Mainz, Germany. [2] Emmy Noether Group for Hominin Meat Consumption, Max Planck Institute for Chemistry, 55128 Mainz, Germany. [3] Institute of Geosciences, Department of Applied and Analytical Paleontology, Johannes Gutenberg University, 55128 Mainz, Germany. [4] Senckenberg Biodiversity and Climate Research Centre, 60325 Frankfurt, Germany. [5] Department of Human Evolution, Max Planck Institute for Evolutionary Anthropology, 04103 Leipzig, Germany. [6] Inorganic Gas Isotope Geochemistry Group, Climate Geochemistry Department, Max Planck Institute for Chemistry, 55128 Mainz, Germany. [7] Department of Heritage, Ministry of Information, Culture and Tourism, 0100 Setthathirath Road, Vientiane Capital, Lao People's Democratic Republic. [8] Université de Paris, BABEL CNRS, UMR 8045 Paris, France. [9] Department of Geosciences, Princeton University, Princeton, NJ 08544, USA. [10] These authors contributed equally: Jennifer N. Leichliter, Tina Lüdecke. ✉email: Jennifer.Leichliter@mpic.de; Tina.Luedecke@mpic.de

D iet is a fundamental driver of evolution, and the development of geochemical proxies that can be used to reconstruct past food webs has been a central focus of paleontological research over the last several decades. While traditional stable isotope analyses of tooth enamel (e.g., carbon and oxygen) and trace element ratios (e.g., Sr/Ca and Ba/Ca) have significantly advanced our understanding of the dietary ecology of fossil organisms[1,2], these proxies do not provide clear information about trophic level. In contrast, the nitrogen (N) isotope ratio of organic material (expressed as $\delta^{15}N$ vs. AIR in ‰, where $\delta^{15}N = [(^{15}N/^{14}N_{sample}/^{15}N/^{14}N_{reference}) - 1] \times 1000$) derived from bulk tissues or specific compounds (e.g., amino acids) is a widely used proxy for the dietary and trophic behavior of animals in terrestrial and marine food webs. More than 40 years of research shows that, in well-constrained systems, consumer tissues are typically enriched by 3 to 4‰ in $^{15}N$ relative to their diet, and $\delta^{15}N$ analyses have been applied across a broad range of ecosystems[3-10].

Despite the potential of nitrogen isotopes for trophic reconstruction, their application in the paleontological record has been limited due to the poor preservation of N-containing organic matter. Nitrogen isotope studies of fossil vertebrates have largely been restricted to collagen extracted from relatively young (e.g., <100 ka), well-preserved bone or dentin samples[11-14]. While recent efforts to develop new trophic proxies have expanded the geochemical tools at our disposal (e.g., to include calcium and zinc isotopes), data documenting baseline variation and variability in the isotopic fractionation of these elements across taxa and tissue types is limited in comparison to the data available for N isotopes[15,16].

Enamel, bone, and dentin—the three phosphatic hard tissues most often preserved in the fossil record—are composed of an inorganic bioapatite fraction (hydrated calcium phosphate mineral with structural and adsorbed carbonate ions; $Ca_{10}(PO_4)_6(OH)_2$)) and an organic matrix (i.e., proteins and lipids that vary in composition depending on the tissue)[17-20]. Relative to enamel, bone and dentin are poorly mineralized (~60 to 70% wt.), with smaller bioapatite crystals and a higher organic fraction (30%) made up predominantly of collagen. The organic matter in bone and dentin is, as a result, susceptible to diagenetic alteration, particularly in unfavorable depositional contexts (e.g., humid environments and acidic soils). As such, reliable bulk or compound-specific $\delta^{15}N$ values are rare for fossil material older than tens of thousands of years[21-24]. In contrast, tooth enamel is more resistant to diagenetic alteration and preserves well across long (e.g., million-year) timescales[20,25-30]. The high-density, crystalline structure of mature tooth enamel (~85 to 95% wt.[20]) effectively encloses and protects organic matter within the biomineral matrix itself. Until recently, however, efforts to measure nitrogen isotopes in the organic matter of tooth enamel have been hampered by the low-N content of enamel (~0.5 to 2% wt. depending on taxon[20,27,30,31]) and the concomitantly large sample size that would be required for isotope analysis via traditional combustion with elemental analyzer isotope ratio mass spectrometry (EA-IRMS). While a modified EA-IRMS system known as a nano-EA has been developed[32], analysis of enamel-bound nitrogen isotopes using this system have not been reported. More generally, sample size requirements for the nano-EA-IRMS method are still prohibitive for many applications (i.e., sampling of small and/or precious fossil teeth), and analytical precision remains relatively low. The oxidation-denitrification method[33] drastically improves analytical precision from ~1.0‰ 2σ standard deviation for nano-EA measurements at 25 nmol of N[32,34] to <0.2‰ at 5 nmol of N. Moreover, the rigorous pretreatment protocol of the oxidation-denitrification method,

conducted in a dedicated low-N clean lab, ensures that only endogenous, enamel-bound nitrogen is measured with low N blanks[35].

Leichliter et al.[35] measured nitrogen isotopes of organic matter in the tooth enamel ($\delta^{15}N_{enamel}$) of rodents from a controlled feeding experiment using the oxidation-denitrification method and established that $\delta^{15}N_{enamel}$ records the nitrogen isotope composition of the diet. In the study, experimental animals that received plant-based diets had significantly lower $\delta^{15}N_{enamel}$ values than those fed meat-based diets. Subsequently, Lüdecke and Leichliter et al.[36] showed that the $\delta^{15}N_{enamel}$ values of African mammals living in a single, well-constrained natural ecosystem also reflect trophic level differences (i.e., carnivore $\delta^{15}N_{enamel}$ values were 4.0‰ higher, on average, than those of herbivores). However, the study was limited to a single ecosystem (Gorongosa National Park in Mozambique) and mammalian carnivores were poorly represented ($n = 2$). Moreover, $\delta^{15}N$ values were measured for tooth enamel only and thus could not be directly compared to $\delta^{15}N$ values of collagen (or any other tissue) from the same individuals. To further establish $\delta^{15}N_{enamel}$ as a trophic level proxy and as a new tool for reconstructing ancient food webs, additional data from both modern and fossil fauna from natural ecosystems are needed.

Here we evaluate whether $\delta^{15}N_{enamel}$ records dietary and trophic level information in modern mammals ($n = 54$) from several different ecosystems across Africa (Fig. S1, Table 1, and S1). Herbivores, including browsers (consuming mainly $C_3$-plants), grazers (consuming mainly $C_4$-plants), and mixed feeders (consuming both $C_3$- and $C_4$-plants), as well as omnivores and carnivores, were selected to ensure that different trophic levels and feeding behaviors were represented. In addition, paired $\delta^{15}N_{bone-collagen}$ values were measured for a subset ($n = 33$) of the same individuals to evaluate whether enamel and bone collagen (the material in which $\delta^{15}N$ has most often been measured in archeological and fossil contexts[37]) record similar dietary information.

Recent evidence from Martínez-García et al.[38] demonstrates that the $\delta^{15}N$ values and N contents of modern and fossil tooth enamel remain stable under experimentally induced conditions that favor organic matter degradation, including oxidative attack, partial dissolution, and heating. Taken together, these observations suggest that enamel-bound organic matter is highly protected and therefore has strong potential to preserve in vivo nitrogen isotope compositions even in diagenetically active settings. Further evidence for this comes from the work of Kast et al.[39], who successfully reconstructed the trophic behavior of extinct megatooth sharks across the Cenozoic (~66 to 3.5 Ma) using $\delta^{15}N$ values obtained from analysis of the enameloid of fossil shark teeth. It remains an open question, however, if fossil mammalian enamel preserves diet and trophic level information in a similar manner to the enameloid of shark teeth, given both the differences in mineral structures between mammalian enamel (hydroxylapatite) and shark enameloid (fluorapatite) as well as the differences in depositional environments. To date, tooth enamel $\delta^{15}N$ values have not been measured in fossil mammalian ecosystems.

Here we apply the oxidation-denitrification method to fossil mammalian teeth from a Late Pleistocene (38.4 to 13.4 ka) terrestrial assemblage from Tam Hay Marklot Cave (THM), Laos in southeast (SE) Asia (Fig. S2). Fossils from this site are characterized by excellent enamel preservation, but poor preservation of bone and dentin, thus precluding $\delta^{15}N$ analysis of collagen for most samples[40]. Additionally, Bourgon et al.[40] measured zinc isotopes in the same specimens to reconstruct the trophic level. As such, the THM fauna represents an ideal test for the

**Table 1 Summary data for modern African mammal specimens ($n = 54$) including diet, common name, species attribution, sample ID, carbon and nitrogen isotope values in ‰ with mean values and the number of individuals ($n =$ ) grouped by diet, and nitrogen content in nmol/mg.**

| Diet | Common name | Taxon | Sample ID | Locality | Water-dependence | Tooth | $\delta^{13}C_{bone\text{-}collagen}$ (‰ vs. VPDB) | $\delta^{13}C_{enamel}$ (‰ vs. VPDB) | $\delta^{15}N_{bone\text{-}collagen}$ (‰ vs. AIR) | $\delta^{15}N_{enamel}$ (‰ vs. AIR) | N content (nmol/mg) |
|---|---|---|---|---|---|---|---|---|---|---|---|
| Browser | Blue Diker | *Philantomba monticola* | 4484 | Dondo, Angola | None | M3 | −21.3 ± 0.1 (2) | −13.3 ± 0.6 (2) | 7.8 ± 0.1 (2) | 7.4 ± 0.1 (2) | 3.7 ± 0.1 (2) |
| | Blue Diker | *Philantomba monticola* | 4483 | Dondo, Angola | None | M3 | −21.5 ± 0.0 (2) | −12.9 ± 0.6 (2) | 5.3 ± 0.4 (2) | 4.3 ± 0.4 (2) | 5.0 ± 0.2 (2) |
| | Giraffe | *Giraffa camelopardalis* | 1869 | Tanzania | Low | M3 | −20.8 ± 0.0 (2) | −11.8 ± 0.2 (2) | 6.2 ± 0.0 (2) | 7.0 ± 0.3 (2) | 5.5 ± 0.3 (2) |
| | Giraffe | *Giraffa camelopardalis* | 9816 | Wamba, Kenya | Low | M3 | – | −10.1 ± 0.1 (2) | – | 6.4 ± 0.3 (2) | 7.7 ± 1.6 (2) |
| | Gorilla | *Gorilla gorilla* | 7902 | Gabon | High | M3 | – | −15.5 ± 0.1 (2) | – | 5.7 ± 0.1 (2) | 12.3 ± 0.3 (2) |
| | Gorilla | *Gorilla gorilla* | 8192 | Cameroon | High | M3 | – | −15.4 ± 0.2 (2) | – | 6.4 ± 0.2 (2) | 7.8 ± 0.5 (2) |
| | Gorilla | *Gorilla gorilla* | 7113 | Sanga, Congo | High | M3 | – | −14.5 ± 0.3 (2) | – | 4.4 ± 0.9 (2) | 4.6 ± 1.4 (2) |
| | Gorilla | *Gorilla gorilla* | 1857 | Congo | High | M3 | – | −15.9 ± 0.3 (2) | – | 6.8 ± 0.5 (2) | 7.1 ± 0.6 (2) |
| | Greater Kudu | *Tragelaphus strepsiceros* | 4510 | Angola | Low | M3 | – | −11.5 ± 0.4 (2) | – | 5.1 ± 0.4 (2) | 5.3 ± 0.7 (2) |
| | Greater Kudu | *Tragelaphus strepsiceros* | 4509 | Rio Caporello, Angola | Low | M3 | – | −12.5 ± 0.8 (2) | – | 6.3 ± (1) | 4.9 ± (1) |
| | Greater Kudu | *Tragelaphus strepsiceros* | 5648 | Ruacana, Angola | Low | M3 | −19.6 ± 0.0 (2) | −12.0 ± 0.2 (2) | 7.2 ± 0.1 (2) | 5.5 ± 0.6 (2) | 4.0 ± 0.0 (2) |
| | Greater Kudu | *Tragelaphus strepsiceros* | 4508 | Taka, Angola | Low | M3 | – | −11.1 ± (1) | – | 5.9 ± (1) | 4.3 ± (1) |
| | Rhino, Black | *Diceros bicornis* | 2553 | Hluhluwe, South Africa | High | M3 | −20.5 ± 0.0 (2) | −12.6 ± 0.4 (2) | 5.1 ± 0.0 (2) | 5.3 ± 0.5 (2) | 2.1 ± 0.1 (2) |
| | Rhino, Black | *Diceros bicornis* | 1865 | East Africa | High | M3 | −22.2 ± 0.0 (2) | −13.2 ± 0.2 (2) | 4.7 ± 0.2 (2) | 4.7 ± 0.2 (2) | 6.0 ± 1.2 (2) |
| **Browser mean values ($n = 14$)** | | | | | | | **−21.0 ± 0.9 (12)** | **−13.0 ± 1.8 (27)** | **6.1 ± 1.2 (12)** | **5.8 ± 1.0 (26)** | **5.6 ± 2.4 (26)** |
| Grazer | African Buffalo | *Syncerus caffer* | 6773 | Aberdare, Kenya | High | M3 | −12.9 ± 0.0 (2) | −2.9 ± (1) | 6.1 ± 0.1 (2) | 5.0 ± 0.0 (2) | 4.7 ± 0.4 (2) |
| | African Buffalo | *Syncerus caffer* | 5649 | Dirico, Angola | High | M3 | −9.5 ± 0.0 (2) | −0.6 ± 0.1 (2) | 7.2 ± 0.0 (2) | 6.0 ± (1) | 3.7 ± (1) |
| | African Buffalo | *Syncerus caffer* | 3913 | Dondo, Angola | High | M3 | −9.2 ± 0.0 (2) | −2.6 ± 0.1 (2) | 4.1 ± 0.1 (2) | 5.8 ± (1) | 4.2 ± (1) |
| | African Buffalo | *Syncerus caffer* | 9566 | Tanzania | High | M3 | −10.4 ± 0.1 (2) | −1.3 ± 0.2 (2) | 8.7 ± 0.2 (2) | 6.8 ± 0.1 (2) | 4.7 ± 0.7 (2) |
| | Hippo | *Hippopotamus amphibius* | 9588 | no information | High | M3 | – | −7.4 ± 0.4 (2) | – | 7.3 ± 0.2 (2) | 3.7 ± 0.2 (2) |
| | Oribi | *Ourebia ourebi* | 7943 | Cambembe, Angola | None | M3 | – | 0.4 ± 0.1 (2) | – | 3.9 ± 0.5 (2) | 6.1 ± 0.1 (2) |
| | Oribi | *Ourebia ourebi* | 7944 | Chana, Angola | High | M3 | −7.9 ± (1) | −0.2 ± 0.4 (2) | 4.5 ± (1) | 4.0 ± 0.2 (2) | 5.6 ± 1.0 (2) |
| | Rhino, White | *Ceratotherium simum* | 2552 | Umfolozi, South Africa | High | M2 | −9.3 ± 0.0 (2) | 0.4 ± 0.4 (2) | 4.7 ± 0.0 (2) | 6.5 ± 1.5 (2) | 3.7 ± 1.2 (2) |
| | Warthog | *Phancochoerus aethiopicus* | 4511 | Cubal, Angola | Low | M3 | −8.2 ± 0.0 (2) | −3.2 ± 0.2 (2) | 6.0 ± 0.0 (2) | 3.5 ± 0.7 (2) | 3.7 ± 0.1 (2) |
| | Warthog | *Phancochoerus aethiopicus* | 4490 | Cubal, Angola | Low | M2 | −8.1 ± 0.1 (2) | −2.6 ± 0.3 (2) | 4.4 ± 0.0 (2) | 4.5 ± 1.2 (2) | 3.4 ± 0.9 (2) |
| | Warthog | *Phancochoerus aethiopicus* | 6739 | Nanyuki, Kenya | Low | M3 | −7.3 ± 0.0 (2) | −1.9 ± 0.2 (2) | 5.0 ± 0.1 (2) | 8.4 ± 0.3 (2) | 2.9 ± 0.9 (2) |
| | Wildebeest, Black | *Connochaetes gnu* | 7938 | no information | High | M3 | – | 1.8 ± 0.2 (2) | – | 6.8 ± (1) | 4.6 ± (1) |
| | Wildebeest, Blue | *Connochaetes taurinus* | 6775 | Kajiado, Kenya | High | M3 | −6.6 ± 0.1 (2) | 1.9 ± 1.3 (2) | 8.2 ± 0.0 (2) | 6.5 ± 0.5 (2) | 3.7 ± 0.2 (2) |
| | Wildebeest, Blue | *Connochaetes taurinus* | 6774 | Kenya | High | M3 | – | 1.8 ± 0.2 (2) | – | 6.5 ± 0.9 (2) | 3.4 ± 0.4 (2) |
| | Wildebeest, Blue | *Connochaetes taurinus* | 6776 | Maasai-Mara, Kenya | High | M3 | −6.8 ± 0.0 (2) | 1.4 ± (1) | 8.5 ± 0.2 (2) | 9.6 ± 0.1 (2) | 3.4 ± 0.0 (2) |
| **Grazer mean values ($n = 15$)** | | | | | | | **−8.7 ± 1.7 (23)** | **−1.0 ± 2.5 (28)** | **6.4 ± 1.9 (23)** | **6.1 ± 1.7 (27)** | **4.1 ± 0.9 (27)** |
| Mixed Feeder | African Elephant | *Loxodonta africana* | 8397 | Cunene, Angola | High | M4-6 | −19.0 ± 0.1 (2) | −10.7 ± 0.2 (2) | 7.9 ± 0.0 (2) | 5.4 ± 1.6 (2) | 2.5 ± 0.0 (2) |
| | African Elephant | *Loxodonta africana* | 8232 | Virunga, Uganda | High | M4-6 | −21.8 ± 0.0 (2) | −12.7 ± 0.3 (2) | 7.6 ± 0.1 (2) | 5.9 ± 0.8 (2) | 4.5 ± 0.9 (2) |
| | African Elephant | *Loxodonta africana* | 8398 | Cunene, Angola | High | M4-6 | −20.4 ± 0.0 (2) | −10.6 ± 0.1 (2) | 12.3 ± 0.0 (2) | 10.0 ± 0.0 (2) | 7.9 ± 5.5 (2) |
| | Impala | *Aepyceros melampus* | 5687 | Secadiva, Angola | High | M3 | – | −4.4 ± 0.2 (2) | – | 6.8 ± 0.2 (2) | 3.1 ± 0.0 (2) |
| | Impala | *Aepyceros melampus* | 5688 | Secadiva, Angola | High | M3 | – | −7.4 ± 0.2 (2) | – | 7.2 ± 0.3 (2) | 7.8 ± 3.1 (2) |
| | Springbok | *Antidorcas marsupialis* | 4539 | Capolopoppo, Angola | Low | M3 | – | −10.6 ± 0.1 (2) | – | 9.2 ± 0.4 (2) | 3.6 ± 0.1 (2) |
| | Springbok | *Antidorcas marsupialis* | 3776 | Capolopoppo, Angola | Low | M3 | – | −8.8 ± 0.5 (2) | – | 8.3 ± 0.7 (2) | 3.4 ± 0.1 (2) |
| **Mixed feeder mean values ($n = 7$)** | | | | | | | **−20.4 ± 1.4 (6)** | **−9.3 ± 2.7 (14)** | **9.3 ± 2.6 (6)** | **7.5 ± 1.7 (14)** | **4.7 ± 2.2 (14)** |
| **Herbivore mean values ($n = 36$)** | | | | | | | **−13.9 ± 6.3 (41)** | **−7.3 ± 6.0 (69)** | **6.7 ± 2.0 (41)** | **6.2 ± 1.6 (67)** | **4.9 ± 2.0 (67)** |
| Omnivore | Baboon, Yellow | *Papio cynocephalus* | 6791 | Lake Baringo, Kenya | High | M3 | – | −10.7 ± 0.4 (2) | – | 7.3 ± 0.2 (2) | 2.4 ± 0.2 (2) |
| | Baboon, Yellow | *Papio cynocephalus* | 6790 | Lake Baringo, Kenya | High | M3 | – | −9.5 ± 0.3 (2) | – | 8.0 ± 0.1 (2) | 2.6 ± 0.1 (2) |
| | Baboon, Yellow | *Papio cynocephalus* | 6795 | Lake Baringo, Kenya | High | M3 | – | −7.5 ± 0.3 (2) | – | 7.2 ± 0.8 (2) | 2.1 ± 0.3 (2) |
| | Baboon | *Papio sp.* | 10960 | Makania, Tanzania | High | M3 | – | −8.0 ± 0.3 (2) | – | 7.2 ± (1) | 7.3 ± (1) |
| **Omnivore mean values ($n = 4$)** | | | | | | | **–** | **−8.9 ± 1.5 (8)** | **–** | **7.4 ± 0.4 (7)** | **3.6 ± 2.5 (7)** |
| Carnivore | African Wild Dog | *Lycaon pictus* | 3797 | Capelongo, Angola | High | M2 | −18.1 ± 0.0 (2) | −11.1 ± 0.2 (2) | 7.8 ± 0.1 (2) | 7.9 ± 0.7 (3) | 7.1 ± 2.2 (3) |
| | African Wild Dog | *Lycaon pictus* | 4678 | Capelongo, Angola | High | M2 | −16.3 ± 0.0 (2) | −11.7 ± 0.2 (2) | 9.6 ± 0.0 (2) | 10.4 ± 0.2 (2) | 9.1 ± 0.2 (2) |
| | Leopard | *Panthera pardus* | 4682 | Luati, Angola | Low | M1 | −13.4 ± 0.7 (2) | −8.8 ± 0.2 (2) | 7.4 ± 0.1 (2) | 7.2 ± 0.5 (2) | 3.9 ± 0.0 (2) |
| | Leopard | *Panthera pardus* | 4683 | Angola | Low | M1 | – | −10.5 ± 0.0 (2) | – | 8.0 ± 0.3 (2) | 5.2 ± 1.6 (2) |
| | Lion | *Panthera leo* | 4677 | Angola | Low | M1 | −12.6 ± 0.0 (2) | −7.5 ± 0.3 (2) | 10.6 ± 0.1 (2) | 9.8 ± 0.6 (3) | 4.8 ± 1.4 (3) |
| | Lion | *Panthera leo* | 5178 | Angola | Low | M1 | −11.6 ± 0.0 (2) | −6.9 ± 0.1 (2) | 11.0 ± 0.0 (2) | 9.8 ± 0.6 (3) | 4.1 ± 0.7 (3) |
| | Lion | *Panthera leo* | 8028 | Dondo, Angola | Low | M1 | −6.8 ± 0.0 (2) | −4.5 ± (1) | 9.4 ± 0.1 (2) | 8.4 ± (1) | 3.9 ± (1) |
| | Lion | *Panthera leo* | 8666 | Etosha Pan, Namibia | Low | M1 | −9.8 ± 0.0 (2) | −4.4 ± (1) | 12.6 ± 0.0 (2) | 9.2 ± 1.1 (2) | 5.1 ± 1.9 (2) |
| | Lion | *Panthera leo* | 7888 | Koma-Region, Tanzania | Low | M1 | −6.6 ± 0.1 (2) | −1.7 ± 0.3 (2) | 10.3 ± 0.1 (2) | 10.3 ± 0.0 (2) | 3.9 ± 0.2 (2) |
| | Spotted Hyena | *Crocuta crocuta* | 3236 | Kenya | High | M1 | −7.1 ± 0.0 (2) | −3.1 ± 0.2 (2) | 9.6 ± 0.0 (2) | 11.1 ± 0.1 (2) | 3.4 ± 0.2 (2) |
| | Spotted Hyena | *Crocuta crocuta* | 949 | Uganda | High | M1 | – | −4.3 ± 0.2 (2) | – | 14.9 ± 0.4 (2) | 3.2 ± 0.2 (2) |
| | Spotted Hyena | *Crocuta crocuta* | 5629 | Dirico, Angola | High | M1 | −16.5 ± 0.0 (2) | −11.5 ± 0.1 (2) | 9.8 ± 0.0 (2) | 9.2 ± 1.1 (2) | 3.5 ± 0.4 (2) |
| | Spotted Hyena | *Crocuta crocuta* | 4675 | Dondo, Angola | High | M1 | −10.8 ± 0.1 (2) | −7.0 ± 0.3 (2) | 8.9 ± 0.1 (2) | 10.5 ± 0.1 (2) | 4.2 ± 0.2 (2) |
| | Spotted Hyena | *Crocuta crocuta* | 8034 | Serengeti, Tanzania | High | M1 | −5.5 ± 0.0 (2) | −2.6 ± 0.1 (2) | 11.0 ± 0.0 (2) | 12.4 ± (1) | 9.6 ± (1) |
| **Carnivore mean values ($n = 14$)** | | | | | | | **−11.3 ± 4.3 (24)** | **−6.8 ± 3.5 (26)** | **9.8 ± 1.4 (24)** | **9.9 ± 2.0 (29)** | **5.1 ± 2.1 (29)** |

Water dependency after[100]–[103]. The number of analyses, typically duplicates, are given in brackets after isotope values. Bolded text indicates mean isotope values for browser, grazer, mixed feeder, omnivore, and carnivore diet categories.

application of $\delta^{15}N_{enamel}$ to the fossil record. We evaluate whether fossil enamel N content is in the same range as modern tooth enamel, consistent with good enamel-bound organic matter preservation, and assess whether the $\delta^{15}N_{enamel}$ values of fossil mammalian teeth preserve dietary and trophic level information in a setting where the degradation of collagen prevents analysis of $\delta^{15}N_{collagen}$.

## Results

$\delta^{15}N_{enamel}$ and $\delta^{13}C_{enamel}$ were measured in modern ($n = 54$) and fossil ($n = 10$) teeth (primarily the latest forming molars from adult individuals). In addition, stable isotope values from mandibular collagen ($\delta^{15}N_{bone-collagen}$ and $\delta^{13}C_{bone-collagen}$) were measured in a subset ($n = 33$) of the modern African fauna. Fossil dentin $\delta^{15}N_{dentin-collagen}$ and $\delta^{13}C_{dentin-collagen}$ values ($n = 4$) are from Bourgon et al.[40]. Isotopic data for all samples are reported in Tables 1, 2 and in Supplementary Data 1, 2.

### Modern African mammals

*Nitrogen isotopes.* $\delta^{15}N_{enamel}$ values of modern African mammals ranged from 3.5 to 14.9‰ ($n = 54$) (Fig. 1a, c and Table 1) and differ significantly according to diet ($F(2,51) = 26.05$, $p < 0.001$). Mean $\delta^{15}N_{enamel}$ values were lowest in herbivores ($6.2 \pm 1.6$‰; $n = 36$) and highest in carnivores ($9.9 \pm 2.0$‰; $n = 14$), while omnivores had intermediate values ($7.4 \pm 0.4$‰; $n = 4$). Herbivores and carnivores differed significantly ($p < 0.001$). Differences were also observed between herbivores with different feeding behaviors (i.e., browsing, grazing, and mixed feeding) and the other dietary groups ($F(4,49) = 15.18$, $p < 0.001$). Across ecosystems, browsing taxa typically had the lowest $\delta^{15}N_{enamel}$ values ($5.8 \pm 1.0$‰; $n = 14$), followed by grazing ($6.1 \pm 1.7$‰; $n = 15$) and then mixed feeding taxa ($7.5 \pm 1.7$‰; $n = 7$; Fig. 1). All herbivore dietary groups differed significantly from carnivores ($p < 0.001$ for both grazers and browsers; $p = 0.015$ for mixed feeders).

$\Delta^{15}N_{bone-collagen}$ values ranged from 4.1 to 12.6‰ ($n = 33$) (Fig. 1b, d and Table 1) and differed significantly according to diet ($F(3,29) = 11.66$, $p < 0.001$). As in enamel, mean $\delta^{15}N_{bone-collagen}$ values were lowest in herbivores ($6.7 \pm 2.0$‰; $n = 21$) and highest in carnivores ($9.8 \pm 1.4$‰; $n = 12$), and herbivores and carnivores differed significantly ($p < 0.001$). Among herbivores, $\delta^{15}N_{bone-collagen}$ values were similar to $\delta^{15}N_{enamel}$ values. Browsers had the lowest values ($6.0 \pm 1.2$‰; $n = 6$), followed by grazers ($6.4 \pm 1.9$‰; $n = 12$), and mixed feeders ($9.3 \pm 2.6$‰; $n = 3$). Grazers and browsers differed significantly from carnivores ($p < 0.001$), but, in contrast to the enamel dataset, the mixed feeders did not differ from carnivores ($p = 0.95$).

We found a significant, positive correlation between $\delta^{15}N_{enamel}$ and $\delta^{15}N_{bone-collagen}$ values (Pearson's correlation $r$ (31) = 0.865, $p < 0.001$; Fig. 2). An ordinary least squares regression yielded the following relationship: $\delta^{15}N_{enamel} \sim 0.88$ [95% confidence interval (CI): 0.66 to 1.1] $\times \delta^{15}N_{bone-collagen} + 0.43$‰ [95% confidence interval (CI): −0.8 to 2.0] with no consistent, directional offset in $\delta^{15}N$ values.

Trophic enrichment between herbivores and carnivores ($\Delta^{15}N = \delta^{15}N_{carnivore} - \delta^{15}N_{herbivore}$) is apparent in both datasets but is larger in enamel (3.7‰) compared to bone collagen (3.1‰) (Fig. 1).

*Carbon isotopes.* $\delta^{13}C_{enamel}$ values ranged from −15.9 to +1.9‰ ($n = 54$; Fig. 3 and Table 1) and differed significantly according to diet ($\chi^2(4) = 41$, $p < 0.001$). Mean $\delta^{13}C_{enamel}$ values were −7.3 ± 6.0‰ for herbivores ($n = 36$), and −6.8 ± 3.5‰ for carnivores ($n = 14$). Browsers had the lowest $\delta^{13}C_{enamel}$ values (−13.0 ± 1.8‰; $n = 14$), followed by mixed feeders (−9.3 ± 3.3‰;

**Table 2 Summary data for fossil (Tam Hay Marklot) specimens including diet, common name, species attribution, sample ID, and carbon, nitrogen, and zinc isotope values in ‰ with mean values and number of individuals ($n =$ ) grouped by diet.**

| Diet | Common name | Taxon | SEVA sample ID | Tooth | $\delta^{13}C_{dentin-collagen}$ (‰ vs. VPDB) | $\delta^{13}C_{enamel}$ (‰ vs. VPDB) | $\delta^{15}N_{dentin-collagen}$ (‰ vs. AIR) | $\delta^{15}N_{enamel}$ (‰ vs. AIR) | N content (nmol/mg) | $\delta^{66}Zn_{enamel}$ (‰ vs. JMC) |
|---|---|---|---|---|---|---|---|---|---|---|
| Browsers | Sumatran serow | *Capricornis sumatrensis* | 34493 | M3 | - | −15.2 (1) | - | 8.9 ± 0.1 (2) | 7.4 ± 0.2 (2) | 0.78 ± 0.02 (2) |
| Browsers | Javan rhinoceros | *Rhinoceros sondaicus* | 34556 | M3 | −24.0 (1) | −15.7 ± 0.1 (2) | 6.6 (1) | 7.5 ± 0.3 (2) | 6.5 ± 2.3 (2) | 0.52 ± 0.01 (2) |
| **Browser mean values ($n = 2$)** | | | | | **−24.0 (1)** | **−15.4 ± 0.4 (3)** | **6.6 (1)** | **8.2 + .1.0 (4)** | **7.0 ± 0.7 (4)** | **0.65 ± 0.18 (4)** |
| Grazers | Asian buffalo | *Bubalus bubalis* | 34524 | P2/P3? | - | −0.8 (1) | - | 4.0 ± 0.0 (2) | 9.3 ± 1.3 (2) | 0.81 ± 0.01 (2) |
| Grazers | Bovine indet. | *Bos sp.* | 34527 | P2 | −9.2 (1) | −1.5 (1) | 3.2 (1) | 2.6 ± 0.1(2) | 7.6 ± 0.2(2) | 0.77 (1) |
| **Grazer mean values ($n = 2$)** | | | | | **−9.2 (1)** | **−1.2 ± 0.5 (2)** | **3.2 (1)** | **3.3 ± 1.0 (4)** | **8.5 ± 1.2 (4)** | **0.79 ± 0.03 (3)** |
| Mixed feeders | Indian muntjac | *Muntiacus muntjak* | 34517 | M3 | −21.0 (1) | −12.9 ± 0.0 (2) | 10.6 (1) | 7.6 ± 0.4 (2) | 5.9 ± 0.2 (2) | 0.29 ± 0.10 (2 |
| **Mixed feeder mean values ($n = 1$)** | | | | | **21.0 (1)** | **−12.9 (2)** | **10.6 (1)** | **7.6 (2)** | **5.9 (2)** | **0.29 (2)** |
| **Herbivore mean values ($n = 5$)** | | | | | **−18.1 ± 7.8 (3)** | **−9.2 ± 7.4 (7)** | **7.1 ± 3.1 (3)** | **6.1 ± 2.7 (10)** | **7.3 ± 1.3 (10)** | **0.63 ± 0.22 (7)** |
| Omnivores | Asian black bear | *Ursus thibetanus* | 34501 | M2 | - | 13.3 (1) | - | 8.8 ± 0.1 (2) | 5.3 ± 0.2 (2) | 0.37 ± 0.02 (2) |
| Omnivores | Wild boar | *Sus cf. scrofa* | 34537 | P4 | −21.7 (1) | 13.3 ± 0.0 (2) | 8.1 (1) | 4.9 ± 0.0 (2) | 4.4 ± 0.0 (2) | 0.32 ± 0.08 (2) |
| Omnivores | Wild boar | *Sus cf. scrofa* | 34538 | P4 | - | 13.7 (1) | - | 5.8 ± 0.6 (2) | 4.4 ± 0.8 (2) | 0.61 ± 0.03 (2) |
| Omnivores | Macaque | *Macaca sp.* | 34548 | M1/M2? | - | 14.3 (1) | - | 9.9 ± 0.1 (2) | 9.9 ± 0.1 (2) | 0.15 ± 0.03 (2) |
| **Omnivore mean values ($n = 4$)** | | | | | **−21.7 (1)** | **−13.6 ± 0.5 (5)** | **8.1 (1)** | **7.3 ± 2.4 (8)** | **5.4 ± 1.5 (8)** | **0.36 ± 0.19 (8)** |
| Carnivores | Asian leopard | *Panthera pardus* | 34505 | P4 | - | −13.4 (1) | - | 11.1 ± 0.5 (2) | 6.1 ± 0.4 (2) | 0.08 ± 0.02 (2) |
| **Carnivore mean values ($n = 1$)** | | | | | **-** | **−13.4 (1)** | **-** | **11.1 (2)** | **6.1 (12** | **0.08 (2)** |

Zinc values are taken from[40]. All individuals were adults. The number of analyses is given in brackets after isotope values. Bolded text indicates mean isotope values for browser, grazer, mixed feeder, omnivore, and carnivore diet categories.

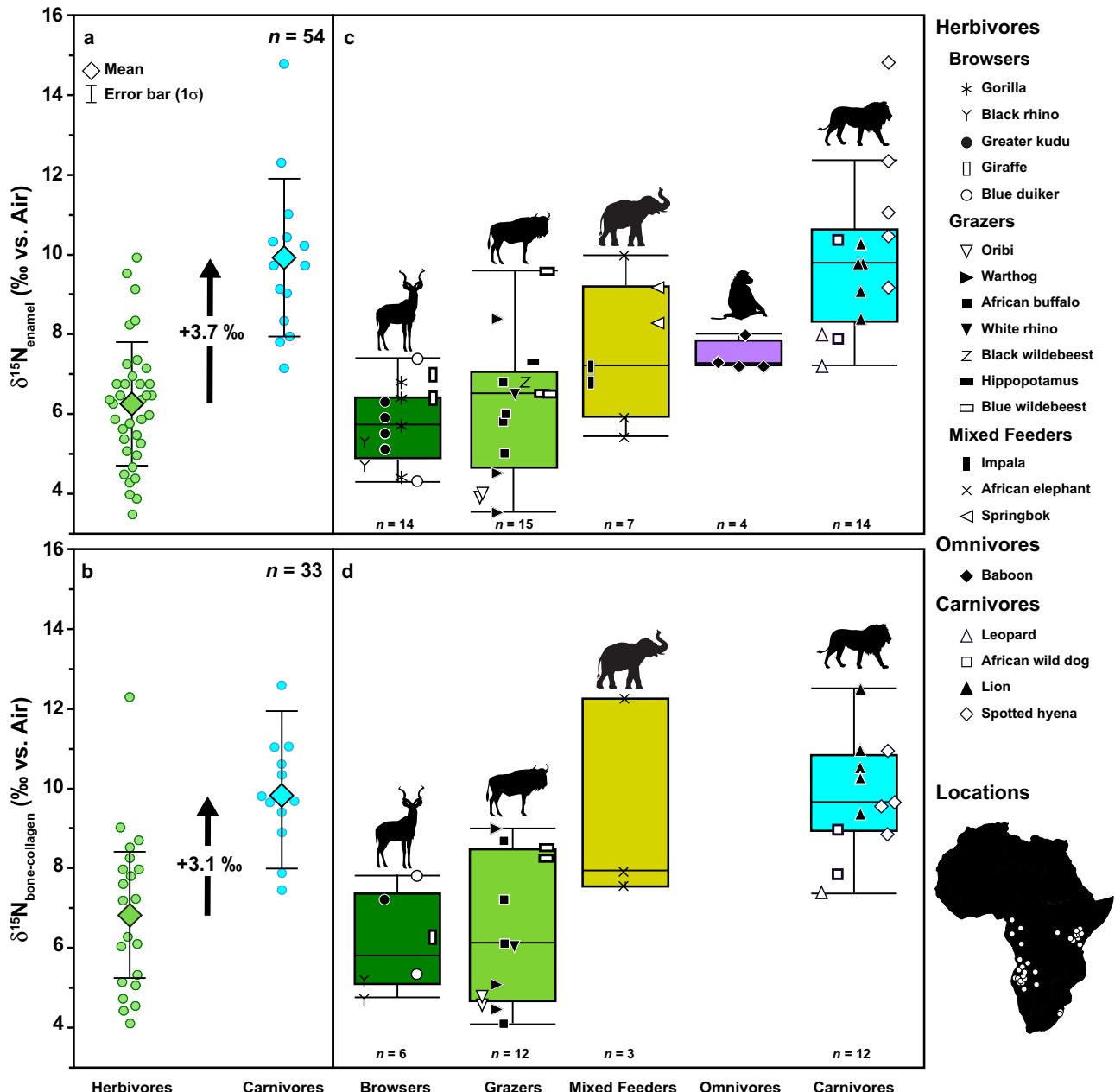

**Fig. 1 Nitrogen isotope ratios measured in enamel (top) and bone collagen (bottom). a** $\delta^{15}N_{enamel}$ and **b** $\delta^{15}N_{bone-collagen}$ values for all modern African herbivores and carnivores measured in this study, with mean and standard deviation ($1\sigma$) indicated. The bolded black arrow illustrates the average trophic enrichment between herbivores and carnivores, which is similar in both datasets. Box plots of **c** $\delta^{15}N_{enamel}$ and **d** $\delta^{15}N_{bone-collagen}$ values for all dietary groups indicating the median and $25^{th}$ percentiles. The $\delta^{15}N$ values for all individuals are plotted by taxon within each dietary group (see Figure legend).

$n = 7$) and then grazers ($-1.0 \pm 2.5\text{‰}$; $n = 15$). Omnivores ($-8.9 \pm 1.5\text{‰}$; $n = 4$) had intermediate values. Carbonate content was typically 4 to 7% wt.

$\delta^{13}C_{bone-collagen}$ values ranged from $-22.2$ to $-5.5\text{‰}$ ($n = 33$; Fig. S4). Mean $\delta^{13}C_{bone-collagen}$ values were $-13.9 \pm 6.3\text{‰}$ ($n = 21$) for herbivores and $-11.2 \pm 4.3\text{‰}$ ($n = 12$) for carnivores. Browsers and mixed feeders had low $\delta^{13}C_{bone-collagen}$ values ($-21.0 \pm 0.9\text{‰}$; $n = 6$ and $-20.4 \pm 1.4\text{‰}$; $n = 3$, respectively), and grazers had significantly higher ones ($-8.7 \pm 1.7\text{‰}$; $n = 12$).

$\delta^{13}C_{enamel}$ was positively correlated with $\delta^{13}C_{bone-collagen}$ (Spearman's correlation $R_s(31) = 0.867$, $p < 0.001$). The positive correlation between $\delta^{13}C_{enamel}$ and $\delta^{13}C_{bone-collagen}$ was stronger within each dietary group (Spearman's correlation $R_s(19) = 0.893$, $p < 0.001$ for herbivores and $R_s(10) = 0.937$, $p < 0.001$ for

carnivores, respectively; see Fig. S4). The offset between $\delta^{13}C_{enamel}$ and $\delta^{13}C_{bone-collagen}$ was higher ($8.2\text{‰}$) in herbivores than in carnivores ($4.5\text{‰}$) as anticipated (see Supplementary Discussion).

## Fossil mammals

*Nitrogen content.* The nitrogen content of modern African mammalian enamel was typically 2 to 10 nmol/mg ($\bar{x} = 4.8 \pm 2.0$ nmol/mg; $n = 54$) after reductive-oxidative cleaning (see Methods). Fossil tooth enamel N content was between 4 and 10 nmol/mg ($\bar{x} = 6.4 \pm 1.6$ nmol/mg; $n = 10$) after cleaning, thus falling within the range of the modern enamel samples (Fig. 4 and Tables 1, 2). No differences were observed between dietary or taxonomic groups in

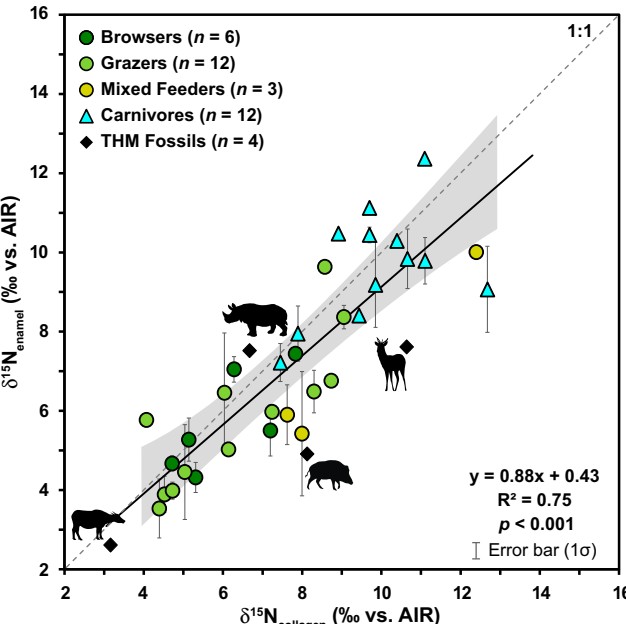

**Fig. 2 Regression of paired δ15Nenamel *versus* δ15Nbone-collagen values ($\bar{x} \pm 1\sigma$) for all modern African mammals ($n = 33$).** The dashed line represents a hypothetical 1:1 relationship between bone collagen and enamel values. The solid line indicates the regression, with a bootstrapped 95% confidence interval illustrated with gray shading. The equation for the regression, $R^2$ value, and $p$ value are indicated in the lower right of the figure. δ15Nenamel *versus* δ15Ndentin-collagen values for the four fossil Tam Hay Marklot (THM) specimens (see Table 2) are also plotted for reference (black diamonds) but were not included in the regression calculation.

either modern or fossil datasets (Fig. S3), and, importantly, no significant correlation was observed between δ15Nenamel and N content (Pearson's correlation $r$ (62) = 0.0033, $p = 0.979$).

*Nitrogen isotopes.* The Late Pleistocene mammals from SE Asia exhibited a range of δ15Nenamel values (2.6 to 11.1‰, $n = 10$) similar to those of the modern African mammals. When added to the regression of the modern African mammal δ15Nbone-collagen and δ15Nenamel values, the four fossil specimens show a moderate, positive correlation, and—although this relationship is not significant (Kendall's Tau $r_\tau = 0.667$, $p = 0.333$)—the regression residuals are mostly within the range of modern mammals (Fig. 2 and S7). As observed in the modern dataset, δ15Nenamel values were lowest in the herbivores (6.1 ± 2.7‰; $n = 5$), highest in the carnivore (a leopard, *Panthera pardus*; 11.1‰), and intermediate in the omnivores (7.3 ± 2.4‰; $n = 4$; Fig. 5). The grazers, including an Asian water buffalo (*Bubalus bubalis*) and an unspecified bovine (*Bos* sp.), had the lowest values (4.0 and 2.6‰, respectively), while the mixed-feeding cervid (*Muntiacus muntjak*; 7.6‰), the browsing rhinoceros (*Rhinoceros sondaicus*; 7.5‰), and the goat-like Sumatran serow (*Capricornis sumatraensis*; 8.9‰) all had higher values. Among the omnivores, the two pigs (*Sus scrofa*; 5.4 ± 0.6‰) had the lowest values, while the macaque (*Macaca* sp.; 9.9‰) and the Asian black bear (*Ursus thibetanus*; 8.8‰) had higher values. Overall, the range of δ15Nenamel values for omnivores encompasses that of herbivores, and thus there are no significant differences between these two diet groups ($\chi^2(2) = 2.9$, $p = 0.235$).

Paired δ15Nenamel and δ66Znenamel values (data from Bourgon et al.[40]) for Tam Hay Marklot ($n = 10$) are negatively correlated (Pearson's correlation $r(8) = 0.671$, $p = 0.034$; Fig. 6).

*Carbon isotopes.* Fossil δ13Cenamel values ranged from −15.7 to −0.8‰ (Fig. 5 and see Fig. S5). δ13Cenamel values were the lowest in the rhinoceros (−15.7‰) and serow (−15.2‰). The mixed-feeding muntjac had an intermediate value of −12.9‰. The bovine (indet.) and Asian water buffalo, both grazers, had the two highest values (−1.5 and −0.8‰, respectively). The suids, macaque, bear, and leopard all had similar δ13Cenamel values, ranging between −14.3 and −13.3‰ (see Table 2). As with modern mammals, carbonate content in the fossil enamel samples was typically between 4 and 7% wt.

Only four THM specimens were sufficiently preserved for δ13Cdentin-collagen analysis (data from Bourgon et al.[40]), but they show a positive relationship with δ13Cenamel data from the same specimens, consistent with the regression obtained for modern specimens (Fig. S4).

## Discussion

Tooth enamel nitrogen and carbon isotopes clearly record diet and trophic level in modern African mammals. We observed an average difference in δ15Nenamel of 3.7‰ between herbivores and carnivores (Fig. 1a). This agrees well with the average trophic enrichment of ~3.5‰ in δ15N documented in numerous large-scale ecological studies[5,10,11]. Moreover, the observed δ15Nenamel values of the different dietary groups—specifically the low values of browsers and grazers, intermediate values of mixed feeders, and high values of carnivores (Fig. 1c)—agree well with published δ15N data for bone collagen[6–8,41,42].

Previous research has shown that both abiotic (e.g., aridity, altitude, and soil chemistry) and biotic (e.g., digestive physiology, protein intake, and water-dependence) factors can cause variation in δ15N between different habitats and within trophic levels, sometimes to such a degree that the overall trophic level effect in nitrogen isotope ratios is obscured[8,41,43–47]. Although the samples included in this study are drawn from multiple localities across Africa (see Fig. S1, Table 1, and Table S1), trophic level patterns are nonetheless clearly discernible in the δ15Nenamel values of animals belonging to different dietary groups. Thus, while regional differences between sampling localities (i.e., baseline variation) may have contributed to observed intra-group variation in δ15Nenamel values, their effects do not obscure the overall trophic level effect recorded in δ15Nenamel.

Additionally, the same individuals' paired δ15Nenamel and mandibular δ15Nbone-collagen values are positively correlated (Fig. 2), confirming that, when collagen is well preserved, enamel and collagen record very similar isotopic information. This finding is important because collagen is the most frequently measured material in nitrogen-based paleodietary studies, and diet-related nitrogen isotope fractionation in bone collagen in African ecosystems is well-understood[4–6]. Thus, demonstrating a clear link between δ15Nenamel and δ15Nbone-collagen represents an important step in establishing δ15Nenamel as a new geochemical approach for reconstructing past diets of fossil vertebrates.

The correlation between bone collagen and enamel δ15N values is not expected to be perfect because there are differences both in when these two hard tissues form as well as in the composition of their organic matter. Bone is a living, dynamic, and actively-growing tissue, which turns over on a time scale of multiple years, incorporating the isotopic composition of an animal's diet throughout its lifetime[48]. In contrast, mammalian teeth mineralize during a discrete period relatively early in an animal's life (i.e., weeks to months or even years, depending on taxon and tooth type and size[49]), after which the enamel becomes metabolically inactive and hence isotopically inert[50]. Thus, the two hard tissues represent different periods in an animal's lifetime and may potentially record different diets; the isotopic

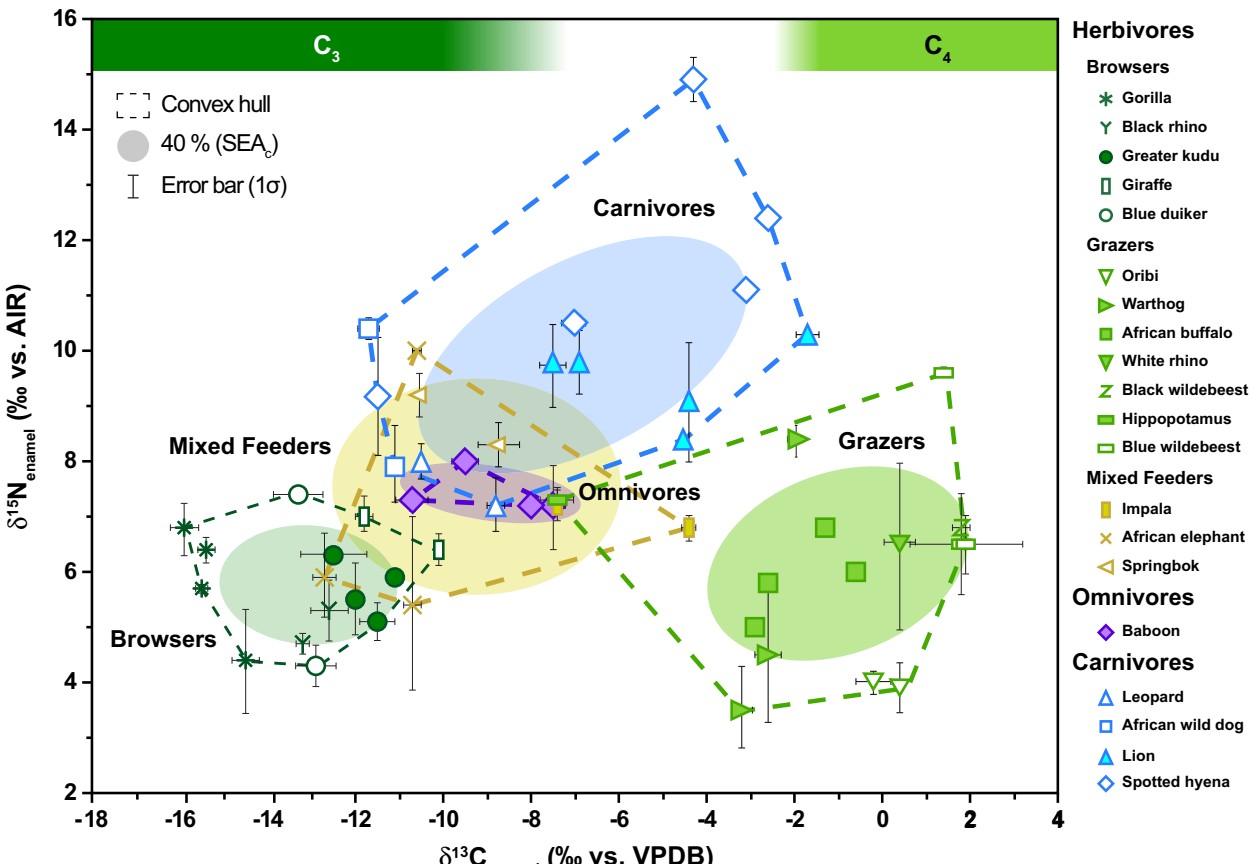

**Fig. 3 Cross plot of $\delta^{13}C_{enamel}$ and $\delta^{15}N_{enamel}$ values for all modern African mammals.** Browsing (dark green), grazing (light green), and mixed feeding (brown-green) herbivores, omnivores (purple), and carnivores (blue) are indicated with color and taxa are indicated with symbols. Dashed lines for convex hulls represent the full range of variation, and shaded ellipses indicate 40% of estimated standard ellipse areas (SEA$_C$). Herbivores and omnivores have relatively low $\delta^{15}N_{enamel}$ values and generally plot in the lower half of the figure, while carnivores have higher $\delta^{15}N_{enamel}$ values and plot in the top half of the figure. Green shaded bars at the top of the figure indicate tooth enamel $\delta^{13}C$ isotopic ranges for modern herbivores corresponding to pure $C_3$ (browsing) and $C_4$ (grazing) resource utilization (after Cerling et al.[104] and corrected for the fossil-fuel-induced shift in the $\delta^{13}C$ of atmospheric $CO_2$ for the period of 1950–1970).

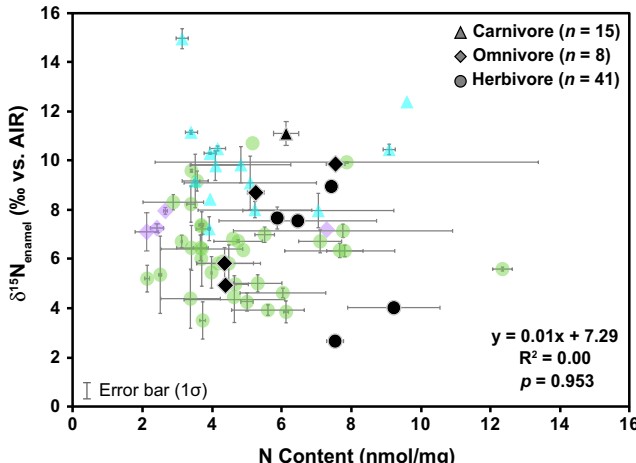

**Fig. 4 $\delta^{15}N_{enamel}$ *versus* N content of modern African mammals (colored symbols) and Tam Hay Marklot fossil mammals (black symbols).** The equation for the regression, $R^2$ value, and *p* value are indicated in the lower right of the figure. Fossil tooth enamel N content falls within the range of modern mammalian tooth enamel N content. $\delta^{15}N_{enamel}$ and N content are not significantly correlated.

composition of mandibular bone usually records the last three to five years before death[48], whereas tooth enamel reflects discrete periods ranging from infancy to juvenile stages, to early adulthood, depending on the tooth type under consideration[49]. In order to capture the adult diet, and avoid enrichment in $^{15}N$ as a result of the consumption of breastmilk[51], we targeted the latest forming tooth in each taxon (typically a molar; see Tables 1, 2).

In addition to different tissue formation times, bone and enamel also differ in the composition of their organic matter. While the organic fraction of bone consists mainly of collagen (90%), the organic matter in enamel is comprised of enamel-specific proteins (predominantly amelogenin) and proteases[27,52–54]. There should thus not necessarily be a perfect 1:1 correlation between $\delta^{15}N_{enamel}$ and $\delta^{15}N_{bone-collagen}$. Nevertheless, the positive correlation between enamel and bone collagen from the same individuals and the clear enrichment in $\delta^{15}N_{enamel}$ across trophic levels confirm that $\delta^{15}N_{enamel}$ records diet and trophic level in a manner similar to $\delta^{15}N_{bone-collagen}$.

Importantly, our dataset also includes measurements of nitrogen and carbon isotope values using the same aliquot of tooth enamel. $\delta^{15}N_{enamel}$ and $\delta^{13}C_{enamel}$ values are plotted together in Fig. 3 and reveal clear, interpretable patterns that are in good agreement with expected dietary habits for the modern taxa

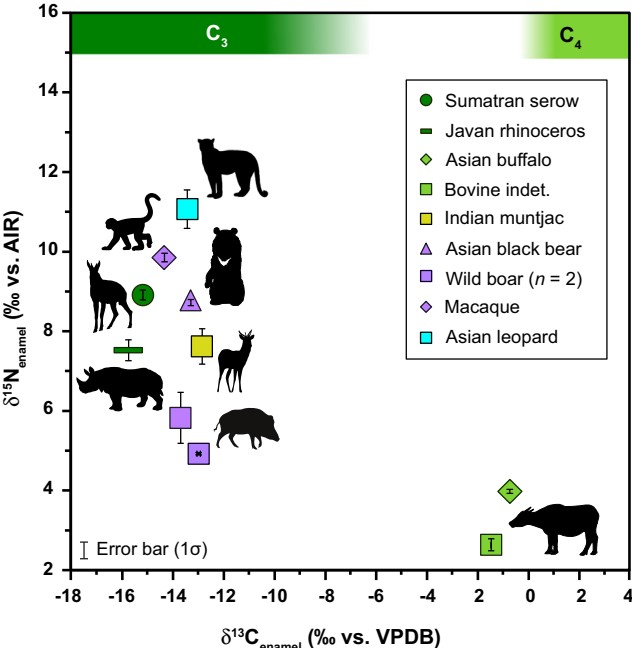

**Fig. 5 Cross plot of $\delta^{13}C_{enamel}$ and $\delta^{15}N_{enamel}$ for all fossil tooth samples from Tam Hay Marklot (THM) Cave.** Herbivore (browsers, dark green; grazers, light green; mixed feeder, brown-green), omnivore (purple), and carnivore (blue) taxa are indicated. One individual was measured for all taxa, with the exception of the wild boar. Green shaded bars at the top of the figure indicate tooth enamel $\delta^{13}C$ isotopic ranges for herbivores corresponding to pure $C_3$ (browsing) and $C_4$ (grazing) resource utilization (enamel $\delta^{13}C$ after Cerling et al.[104] and corrected for a fossil-fuel-induced shift in the $\delta^{13}C$ atmospheric $CO_2$ of ~ −2.0‰[98]). Grazing taxa are clearly distinguished from the rest of the fauna by their higher $\delta^{13}C_{enamel}$ values. Most of the THM fauna lived and foraged in predominantly $C_3$ habitats. Herbivores and omnivores have lower $\delta^{15}N_{enamel}$ values compared to the single carnivore, which has the highest value (*Panthera pardus*).

included in this study. Specific results for herbivores, omnivores, and carnivores are discussed in the following text.

While herbivores have a lower overall mean $\delta^{15}N_{enamel}$ value than carnivores, they exhibit a high degree of variation in $\delta^{15}N_{enamel}$ as a dietary group. Numerous factors have been proposed to drive this variation. Abiotic and biotic variables, including environment (e.g., precipitation, temperature, soil chemistry, and their effect on soil N cycling), physiology (e.g., water conservation and digestive physiology[46,47]), and diet (e.g., feeding strategy, foraging micro-habitat, and macronutrient composition of diet)[8,41,43,55–61] can all influence $\delta^{15}N$. For instance, consumed plant part (e.g., leaf, fruit, stem, and root) and plant nutritional quality (e.g., protein-, fiber-, and fat-content), as well as the digestive anatomy (ruminant vs. non-ruminant) and the nutritional status (e.g., starvation, pregnancy, and lactation) of the animal itself, also can impact herbivore $\delta^{15}N$ values[47,57–65]. Ultimately, the $\delta^{15}N_{enamel}$ values of herbivore body tissues are driven primarily by the isotopic composition of the plants they consume, which is in turn controlled by abiotic factors specific to each ecosystem[55] (see Fig. S6 and Supplementary Discussion for more details).

When grouped according to feeding behavior, the mixed feeders stand out in particular amongst the herbivores. While browsing and grazing taxa do not differ in $\delta^{15}N_{enamel}$, mixed feeders (springbok, impala, and elephants) had significantly higher $\delta^{15}N_{enamel}$ values, even overlapping with those of the carnivores in some cases (Fig. 1c). Despite this overlap, mixed feeders still differ significantly from carnivores in $\delta^{15}N_{enamel}$. This is not the case for

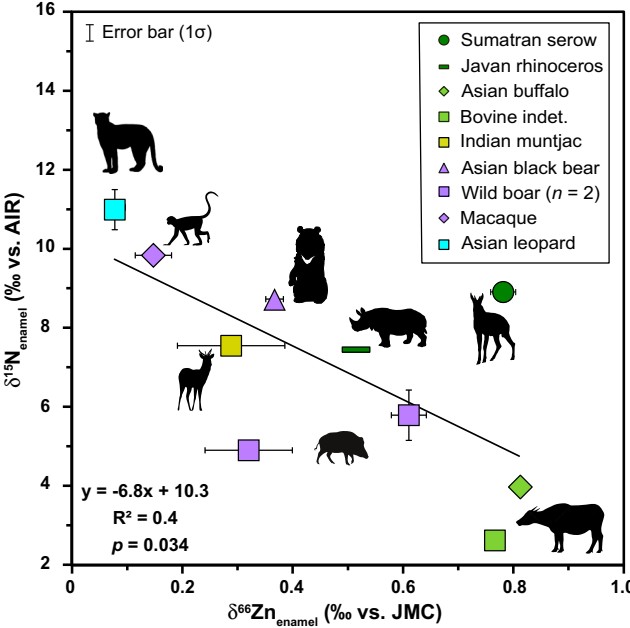

**Fig. 6 Regression of paired $\delta^{15}N_{enamel}$ (this study) *versus* $\delta^{66}Zn_{enamel}$ (from Bourgon et al.[40]) values for the Late Pleistocene fossil teeth from Tam Hay Marklot ($\bar{x} \pm 1\sigma$; $n = 10$).** The black line indicates the regression for all fossils, and the equation for the regression, $R^2$ value, and $p$ value are indicated in the upper right of the figure. The two trophic level proxies are negatively correlated, as expected.

$\delta^{15}N_{bone-collagen}$, although this difference may have been driven, at least in part, by the smaller size of the collagen dataset.

These herbivore $\delta^{15}N$ values are consistent with previously published herbivore data from modern ecosystems[6,8,25,36,41–43,61,66]. Ambrose[41] observed relatively high $\delta^{15}N_{bone-collagen}$ values (also overlapping with those of co-occurring carnivores) in mixed-feeding herbivores from eastern Africa, Sealy et al.[8] observed higher $\delta^{15}N_{bone-collagen}$ values in mixed-feeding springbok compared to other herbivorous taxa in southern Africa, and Codron et al.[42] measured the highest herbivore $\delta^{15}N$ values in the feces of mixed feeders (impala and nyala) in their study of herbivores in Kruger National Park. African elephant tissue $\delta^{15}N$ values vary widely, depending on which region of Africa the animals inhabit, but our observed $\delta^{15}N_{enamel}$ values (5.4 to 10.0‰) fit well within the large range of published bone collagen values (2.0 to 16.0‰) for this taxon[8,67–69]. There is much debate regarding the possible mechanisms driving elevated values in mixed feeders and it remains unclear why mixed feeders tend to have higher $\delta^{15}N$ values than other herbivores overall (see Fig. S6 and Supplementary Discussion for a more detailed discussion of this topic). Our $\delta^{15}N_{enamel}$ dataset for one given locality in Africa is too small to test if this holds true for tooth enamel amongst mixed feeders more broadly. Regardless the overall pattern observed in our dataset is consistent with data published for herbivores from African and South American ecosystems[36,61,66].

For the omnivores in this study, we analyzed enamel from baboons (*Papio*), which are of particular interest for the potential application of this method to questions about human evolution, as baboons live in open savanna ecosystems and have been proposed as a model taxon for early hominins. Baboons are ecological and dietary generalists that consume a wide variety of foods in an opportunistic manner[70–73], including many types of plants, as well as insects, small animals, eggs, etc. However, although baboons are considered dietary generalists, their diet often consists primarily of plant matter[74].

The $\delta^{15}N_{enamel}$ values we obtained for baboons are consistent with existing stable isotope data for these primates[62]. Baboon $\delta^{15}N_{enamel}$ values were significantly lower than those of carnivores and were instead comparable to those of herbivores, specifically the mixed feeders (Fig. 1). Although some studies have found that baboons have low $\delta^{15}N$ values compared to sympatric herbivores[6,62], perhaps related to the consumption of $N_2$-fixing plants and/or underground storage organs, no clear tendency towards lower values was observed in baboon $\delta^{15}N_{enamel}$ values compared to herbivores in our dataset. The teeth sampled for this study would have formed just at the end of the juvenile period, and young, low-ranking baboons rarely consume meat, a behavior that has primarily been observed in dominant adult males[36,75,76]. Thus the relatively low $\delta^{15}N_{enamel}$ values for the baboons are expected.

$\delta^{15}N_{enamel}$ values were the highest for the carnivores, evidencing clear trophic enrichment compared to herbivores and falling into the range of $\delta^{15}N_{bone-collagen}$ values typical for carnivores. Within the carnivore guild, we also observed differences in $\delta^{15}N_{enamel}$ between taxa. Leopards (*Panthera pardus*) and wild dogs (*Lycaon pictus*), for instance, had lower $\delta^{15}N_{enamel}$ values than spotted hyenas (*Crocuta crocuta*) and lions (*Panthera leo*). Overall, the $\delta^{15}N_{enamel}$ values of spotted hyenas are higher than those of other carnivore species, and, in two individuals, substantially higher than all other $\delta^{15}N$ values measured for this species in this study. It is probable that these higher values incorporate a nursing signal, as the permanent dentition of spotted hyenas mineralizes early and erupts at approximately the same time that weaning occurs[77]. If these two outliers are excluded from the carnivore $\delta^{15}N_{enamel}$ dataset, $\Delta^{15}N_{carnivore-herbivore}$ decreases to 3.1 ‰, but trophic enrichment between herbivores and carnivores nevertheless remains significant and equal to that of $\delta^{15}N_{bone-collagen}$.

Although the observed differences between carnivore $\delta^{15}N_{enamel}$ values are potentially related to niche separation and differential habitat use/prey preference, no strong conclusions can be drawn from the differences in $\delta^{15}N_{enamel}$ values alone, especially when considering that these samples are drawn from disparate localities. However, carbon isotope data help shed further light on niche separation between carnivore taxa. For instance, while all carnivores included in this study are typical savanna dwellers, $\delta^{13}C_{enamel}$ values for leopards and wild dogs in this dataset indicate that these two taxa relied more heavily on browsing prey (i.e., $C_3$-consuming prey) compared to spotted hyenas and lions (Fig. 3). Most carnivores selectively hunt specific taxa and their diets can be strongly influenced by competition with other carnivores. Leopards, for example, prefer relatively small prey (e.g., body mass 10 to 40 kg), which occur in dense habitats; for example, impala, bushbuck, and common duiker, while larger prey and species restricted to open vegetation are generally avoided[78]. Similarly, wild dogs tend to hunt in areas of denser vegetation and target browsing and mixed-feeding herbivores in regions where they co-occur with hyenas and lions[79]. Lions and hyenas, in contrast, have higher $\delta^{13}C_{enamel}$ values, consistent with the consumption of a greater proportion of grazing taxa (i.e., $C_4$-consuming prey) which occupy more open environments. These carnivore data illustrate the potential to refine reconstructions of trophic niches using combined $\delta^{15}N_{enamel}$ and $\delta^{13}C_{enamel}$ analyses.

Importantly, this study was not designed to explicitly test the effect of the potential confounding abiotic or biotic factors that may be driving variation in herbivore/carnivore $\delta^{15}N$ values, and our $\delta^{15}N_{enamel}$ dataset for any one given locality in Africa is too small to do so properly. Rather, we set out to demonstrate that $\delta^{15}N_{enamel}$ values record diet in a robust manner comparable to other commonly measured tissue types, especially collagen, the material most frequently measured in the archeological and fossil record. Our data conclusively demonstrate that tooth enamel organic matter records dietary information, confirming the utility of $\delta^{15}N_{enamel}$ as a trophic level proxy.

Tooth enamel isotopes measured in Late Pleistocene fossil teeth indicate good preservation of enamel-bound organic matter and record diet and trophic level information. In their study of zinc isotopes in fossil tooth enamel from THM, Bourgon et al.[40] attempted to extract collagen from dentin for $\delta^{13}C$ and $\delta^{15}N$ analysis. Of 72 samples, 23 had enough dentin to attempt collagen extraction, but only four specimens had sufficiently preserved collagen (i.e., with a C:N between 2.9 and 3.6[14,80]) for stable isotope analysis. Collagen yield was also relatively poor (<1% for all samples). Thus, reconstruction of trophic levels based $\delta^{15}N_{collagen}$ values was not feasible for this assemblage. In contrast, all available THM fossil teeth analyzed using the oxidation-denitrification method ($n = 10$) have nitrogen contents that are comparable to modern tooth enamel (e.g., Leichliter et al.[36], Lüdecke and Leichliter et al.[35], this study; Fig. 4). If additional exogenous N would have been added during fossilization, we would expect fossil samples to have a higher N content than their modern counterparts. Alternatively, if the organic matter was degraded over time, we would expect a clear decrease in N content with respect to modern samples[38,81]. Either scenario, i.e., the addition of exogenous N or degradation of endogenous organic matter, could result in corresponding directional changes in $\delta^{15}N_{enamel}$ values in relation to N content. However, the N content and $\delta^{15}N_{enamel}$ values of the fossil samples are within the range of modern specimens and show no correlation (see Fig. 4), indicating good enamel-bound organic matter preservation in our fossil dataset. These observations are consistent with the results of laboratory degradation experiments[38] and with measurements of million-year-old marine microfossils[39,82,83], which suggest that biomineral structures act as an effective physical barrier that protects organic matter from degradation. Although it is a small dataset, the paired $\delta^{15}N_{enamel}$ and $\delta^{15}N_{dentin-collagen}$ values for the four fossil specimens are also positively, though not significantly, correlated (see Fig. 2 and Fig. S7).

Within the THM assemblage, fossil $\delta^{15}N_{enamel}$ values record trophic enrichment in $^{15}N$ (i.e., low values for herbivores, intermediate values for omnivores, and a high value for the carnivore; Fig. 5). While the current fossil dataset is too small to draw any strong conclusions regarding the trophic structure of the THM fossil assemblage, these patterns corroborate the good preservation of dietary N-isotope compositions in tooth enamel for samples in which the collagen is already degraded. Additionally, we observed some interesting results. For example, the $\delta^{15}N_{enamel}$ values of the browsers were relatively high compared to those of the grazers, deviating from the pattern of low $\delta^{15}N_{enamel}$ values observed for browsers in the modern African fauna. African herbivores living in forests generally exhibit lower $\delta^{15}N_{collagen}$ values than herbivores from more open (i.e., grassland) environments[6,41]. It is plausible that the relatively higher $\delta^{15}N_{enamel}$ values of the browsers from THM may be unique to this region and time period, or reflect selective feeding behavior, as plant $\delta^{15}N$ values are known to vary geographically as well as according to plant taxon, part, and position in the forest canopy[84,85] however, it is difficult to discern any cogent pattern with only a few individuals in each feeding category. The $\delta^{13}C_{enamel}$ values reveal that, except for the two grazers, the analyzed fossil taxa from THM lived and foraged in predominantly $C_3$ environments. This is consistent with the ecology of the fauna present in the assemblage based on taxonomy and dental morphology, and with what is known about the environment in this region of Southeast Asia during the Late Pleistocene, which was probably predominantly

forested, albeit not as densely as a closed low-light tropical rainforest[40,86,87].

$\delta^{15}N_{enamel}$ and $\delta^{66}Zn_{enamel}$ values from the same individuals show a negative correlation (Fig. 6). This is expected as $\delta^{66}Zn_{enamel}$ has been shown to decrease with increasing trophic level[40,88–90]. Encouragingly, reconstructed trophic positions for omnivorous taxa based on the two isotope systems are in good agreement. Specifically, $\delta^{15}N_{enamel}$ value(s) for the pigs are low, the bear is intermediate, and the macaque is high, while the inverse is true for $\delta^{66}Zn_{enamel}$. This finding is especially promising given the difficulty of interpreting variations in bulk tissue $\delta^{15}N$ isotope values (such as those obtained from e.g., enamel organic matter and collagen) for taxa with generalist feeding strategies like omnivores and mixed feeders. Future studies that incorporate both $\delta^{15}N_{enamel}$ and $\delta^{66}Zn_{enamel}$ (and other isotope systems) may help us better resolve omnivores' dietary behavior, a task that is particularly challenging given their broad resource use.

In this study, we present paired organic nitrogen and inorganic carbon isotopic values measured in a single aliquot (5 to 7 mg) of tooth enamel. Importantly, our study demonstrates that the $\delta^{15}N_{enamel}$ values of mammals from natural ecosystems record diet and trophic behavior in the same manner as the classical dietary proxy of $\delta^{15}N_{collagen}$. The $\delta^{15}N_{enamel}$ values of carnivores are elevated by 3 to 5‰ compared to those of herbivores and omnivores in both modern and fossil food webs, which is comparable to the well-established 3 to 4‰ enrichment in $\delta^{15}N$ per trophic level along food chains[3–10]. Furthermore, paired $\delta^{15}N_{enamel}$ and $\delta^{15}N_{bone-collagen}$ values from the same individuals are positively correlated. While more studies comparing bone, dentin, and enamel $\delta^{15}N$ from the same individual will be useful in characterizing the exact nature of the relationship between these different tissue types, the results of these analyses conclusively demonstrate that $\delta^{15}N_{enamel}$ and mandibular $\delta^{15}N_{bone-collagen}$ record similar diet and trophic information. Indeed, given what is known about the inherent variability in $\delta^{15}N$ related to abiotic and biotic factors, the coherence of our modern dataset and the consistency of the enrichment in $^{15}N$ between trophic levels in samples drawn from disparate localities is remarkable, and demonstrates that $\delta^{15}N_{enamel}$ is a robust trophic proxy with great potential for application in paleodietary studies.

The results of our analysis of $\delta^{15}N_{enamel}$ of enamel samples from modern and fossil teeth are highly promising. In the archeological and paleontological record, the degradation of organic matter (specifically collagen) is a fundamental limitation[91] restricting measurements to relatively young, well-preserved samples. Therefore, the ability to measure the nitrogen isotope composition of the organic matter preserved in diagenetically robust tooth enamel (for example, at Tam Hay Marklot Cave) has the potential to be used to investigate the trophic ecology of ancient or even extinct animals over time periods far beyond the limit of collagen preservation. For example, dental wear, stone tools, and cut marks on fossil bones associated with early hominins suggest that members of the genus *Australopithecus* may have engaged in meat consumption as early as 3 million years ago[92]. However, no direct geochemical data exists to evaluate this claim, and the inference that *Australopithecus* shaped and used tools to access animal resources prior to the emergence of *Homo* is heavily debated[93]. $\delta^{15}N_{enamel}$ values of early hominins and associated fauna have the potential to shed new light on this debate by providing geochemical evidence for the onset and intensification of animal-resource consumption throughout human evolution.

## Methods

**Experimental design**. Tooth enamel ($n = 54$) and mandibular bone ($n = 33$) were sampled from 20 modern African mammalian taxa housed in the zoological

collection at the University of Hamburg, Germany. We targeted herbivores (including browsers, grazers, and mixed feeders), omnivores, and carnivores to evaluate the effect of trophic level on $\delta^{15}N$ values. A minimum of three individuals were sampled per taxon. Enamel from the last forming tooth was preferentially used whenever possible to avoid the effect of milk consumption (which typically results in ~2 to 3‰ higher $\delta^{15}N$ values in the tissues of nursing individuals compared to the mother's tissues[51]). Specimens were sampled using a hand-held Dremel with a diamond burr tip. For fossil specimens ($n = 10$; 9 taxa) from THM, a chip of tooth enamel was crushed and ground to a fine powder in an agate mortar and pestle. All data and associated information are presented in Tables 1, 2, Supplementary Tables S1–4, and Supplementary Data 1–4.

**Tooth enamel nitrogen isotope measurement**. Tooth enamel samples (5 to 7 mg) were measured for $\delta^{15}N_{enamel}$ in nine analytical batches using the oxidation-denitrification method. $\delta^{15}N_{enamel}$ values of bacterially converted $N_2O$ were measured via gas chromatography-isotope ratio mass spectrometry at the Max Planck Institute for Chemistry (MPIC, Mainz, Germany). The method used to measure $\delta^{15}N$ values of tooth enamel is described only briefly here; for a detailed description, see Leichliter et al.[35] and references therein. The procedure consists of four main steps: (1) 5 to 7 mg of tooth enamel powder is subjected to a reductive-oxidative cleaning to remove exogenous organic matter[94]; (2) samples are demineralized, and all endogenous organic matter (i.e., intra-and inter-crystalline bound N) is oxidized to nitrate using a persulfate oxidizing reagent (0.67–0.70 g of four times re-crystallized potassium persulfate added to 4 ml of 6.25 N NaOH solution in 95 ml Milli-Q water)[33], (3) nitrate is quantitatively converted to $N_2O$ via the 'denitrifier' method[95], and (4) sample-derived $N_2O$ is extracted, and its nitrogen isotopic composition is measured on a custom system online to a Thermo Scientific™ 253 Plus isotope ratio mass spectrometer (IRMS). Isobaric interference with $CO_2$, is minimized by two stages of cryo-isolation and gas chromatography columns as detailed in[95,96]. These automated steps result in the full separation of the $CO_2$ peak from the $N_2O$ peak, which can be seen in each sample chromatogram. International materials (USGS 40, USGS 65, USGS 41, USGS 34, and IAEA-NO-3) and in-house standards (PO-1, LO-1, AG-Lox, Noto-1; see Leichliter et al.[35] for details) are included in every run and each step of this process, allowing us to monitor measurement stability, and evaluate the possibility of matrix-based effects during cleaning. Blank N concentration and $\delta^{15}N$ were measured for each analysis batch, and the sample N content and $\delta^{15}N$ values were corrected using the blank measurements of the associated batch. Samples were measured in duplicate or triplicate (resulting in a total of 115 individual measurements) and in separate batches whenever possible. Blank N content was between 0.3 and 0.6 nmol/ml ($\bar{x} = 0.37 \pm 0.11$ nmol/ml), resulting in an average blank contribution of 3% or less. Inter-batch precision ($\pm 1\sigma$) in $\delta^{15}N$ for in-house standards was 0.4‰ for coral standards ($n = 43$) and 0.4‰ for tooth enamel standards ($n = 41$) across all analysis batches (see Supplementary Data 3).

**Tooth enamel carbon isotope measurement**. Small aliquots (50–100 μg) of untreated enamel powder were measured using high-precision continuous-flow mass spectrometry. $\delta^{13}C_{enamel}$ analyses were performed at the MPIC on a Thermo Delta-V continuous-flow mass spectrometer coupled to a Gasbench II gas preparation system, equipped with a liquid nitrogen cryogenic trap (i.e., the "cold trap method")[97]. Untreated enamel powder (see Fig. S5) was reacted with >99% $H_3PO_4$ for 90 min at 70 °C before the resulting $CO_2$ was introduced to the continuous flow system. Isotope data are calculated by direct comparison to eleven replicates of a tooth enamel standard (AG-Lox; $\delta^{13}C = -11.46 \pm 0.2$‰) analyzed in each batch. Samples were analyzed in a total of seven analytical batches, and samples were measured in duplicate or triplicate (where possible) in different batches. A logarithmic fit through the isotope ratios *versus* peak size for the AG-Lox replicates was used to eliminate fractionation effects due to sample size. After these corrections, the reproducibility of international and in-house carbonate standards (IAEA-603, NBS 18, VICS), as well as sedimentary phosphate (NIST SRM 120c), was generally better than 0.3‰ (1σ) and <0.1‰ (1σ) within batches for AG-Lox (Supplementary Data 4). Sample $\delta^{13}C$ values were corrected for changes in atmospheric $\delta^{13}C$ where appropriate after Hare et al.[98] and corrections are specified in the text and in the relevant figure captions. Carbonate contents were derived from standard *versus* sample total peak area ratios (7.5% structural carbonate content for AG-Lox; after Vonhof et al.[97]).

**Bone collagen carbon and nitrogen isotope measurement**. Bone collagen was extracted according to the protocol described by Richards and Hedges[99]. Up to 150 mg of mandibular bone powder was demineralized with 0.5 M HCl for at least 24 h at 4 °C. Samples were then centrifuged, the supernatant discarded, and the remaining collagen was rinsed three times with deionized water. Afterward, a pH of 2 to 3 was obtained by adding a few drops of 0.5 M hydrochloric acid. The sample was then gelatinized by heating collagen to 70 °C for 48 h. After 48 h, gelatinized collagen was filtered (using 0.55 μm pore size midi-filters), centrifuged, the supernatant was discarded, and the remaining collagen was freeze-dried. Collagen yield was between 5 and 25%.

$\delta^{13}C_{bone-collagen}$ and $\delta^{15}N_{bone-collagen}$ were measured in the same aliquot of collagen at the Institute for Organic Chemistry, Johannes Gutenberg University Mainz, Germany, using an IsoPrime™ High-Performance Stable Isotope Ratio Mass

Spectrometer, GV Instruments. Each run included the following standards: ten Sulfanilamide replicates, three replicates each of IAEA-N1, IAEA-N2, IAEA-CH6, and IAEA-CH7, and two replicates of the bovine liver (NIST SRM 1577). For collagen measurements, 1 to 2 mg of extracted collagen was analyzed in replicate for each specimen. Analytical precision was better than ± 0.25‰ (1σ) for both $\delta^{13}C$ and $\delta^{15}N$.

**Statistics and reproducibility.** Isotopic datasets were evaluated to determine if the data were normally distributed and if the variance was equal between groups using Levene's test. A non-parametric Kruska–Wallace test followed by a Dunn's post hoc test with a Bonferroni correction for pairwise comparisons was used when the requirements for ANOVA were not satisfied; otherwise, analysis of variance (ANOVA; one-tailed) was used to identify statistically significant differences in isotopic values between groups for modern African fauna. Where ANOVA indicated statistical significance, pairwise comparisons were made using a Tukey–Kramer HSD post hoc test to determine which groups differed from one another in their isotope values. Statistically significant relationships between paired isotopic values from the same individuals were determined using Pearson's correlation (two-tailed), Kendall Rank correlation, or Spearman Rank correlation (two-tailed) where appropriate. Statistical analyses were performed using Paleontological Statistics Version 4.09 (PAST4) and JMP®, Version 16 statistical software using an alpha level for the significance of 0.05. Detailed results of pairwise comparisons can be found in Tables S2–S4. Sample sizes and the number of replicates are stated in the Results section as well as in Supplementary Data 1–4.

**Reporting summary.** Further information on research design is available in the Nature Portfolio Reporting Summary linked to this article.

## Data availability

The Supplementary Materials include additional information regarding analytical procedures, data processing, sampling locations, and isotopic data for the samples included in this study. All isotope values are provided in the main text as well in an associated excel database (Supplementary Data 1–4).

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

## Acknowledgements

We thank F. Rubach, S. Brömme, M. Schmitt, and B. Hinnenberg (Climate Geochemistry Department, Max Planck Institute for Chemistry, Germany) for technical support, D. Winkler (Zoological Institute, University of Kiel, Germany) and T. Kaiser (Center of Natural History, University of Hamburg, Germany) for assistance in sampling the modern African fauna. We thank J. Broska and P. Held for collagen preparation and $\delta^{13}C/\delta^{15}N$ analysis of bone collagen. We also thank S. Luangaphay (Department of National Heritage, Ministry of Information and Culture in Vientiane, Laos) for the authorization to study the published fauna of Tam Hay Marklot, P.O. Antoine, who aided in taxonomic identification and analysis of the fauna, and researchers who are part of the Laos project and participated in the fieldwork (F. Demeter, L. Shackelford, P. Duringer, J.L. Ponche, Q. Boesch, E. Patole-Edoumba, T.E. Dunn, A. Zachwieja, E. Suzzoni, S. Frangeul, S. Duangthongchit, T. Sayavonkhamdy, P. Sichanthongtip, and D. Sihanam). This study was funded by the Max Planck Society (A. Martínez-García), the European Research Council (ERC) under the European Union's Horizon 2020 Research and Innovation Program (Grant Agreement 681450) (ERC Consolidator Grant Agreement to T. Tütken); the Paul Crutzen Nobel Prize fellowship of the Max Planck Society to N.N. Duprey; and the Deutsche Forschungsgemeinschaft (DFG) Grant LU 2199/1-2 and the Emmy Noether Fellowship LU 2199/2-1 to T. Lüdecke. Prior method development work was supported by the Scott Fund of the Department of Geosciences, Princeton University (to D.M.S.).

## Author contributions

Author contributions: J.N.L., T.L., T.T., and A.M.-G. designed the research; J.N.L. and T.L. performed enamel N isotope analyses in the laboratory of A.M.G. with assistance from A.D.F. and N.N.D.; T.T. contributed the bone collagen C and N isotope data; J.N.L. and T.L. performed C and O isotope analyses in the laboratory of H.V., J.N.L. and T.L. analyzed the data; A.-M.B. conducted taxonomic identification and analysis of the fauna; V.S., A.M.B., and N.B. represent the LAOS project team; D.M.S., T.T., and A.M.-G. assisted in the interpretation of the data; J.N.L. and T.L. wrote the paper. All authors contributed to the interpretation of the data and provided input to the final manuscript.

## Funding

## Competing interests

The authors declare no competing interests.
