## [Peer Review File · Communications Biology]

Reviewers' comments:

Reviewer #1 (Remarks to the Author):

Review of the manuscript "Nitrogen isotopic composition of tooth enamel organic matter records trophic position in modern and fossil ecosystems" by Leichter et al. for Communications Biology.

This is an interesting paper that pushes analytical limits of measuring N isotopes in modern and fossil enamel. I have no major concerns and would recommend publication providing minor revisions.

The main point is that I found the first paragraph of the introduction a bit clumsy. Regarding Ca and Zn isotopes, the authors write that "data documenting baseline variation ... is limited. As such, a reliable, well-characterized geochemical proxy for determining trophic position in deep time remains elusive." This is clearly not the case for Ca isotopes which are particularly efficient for very old trophic systems. Thomas Tütken must know about this as he co-authored one important paper on Ca isotopes in dinosaurs. The reason why Ca isotopes are very efficient at reconstructing trophic relationships is when bone comes with the whole prey, which is not necessarily the case regarding mammals. The authors should acknowledge this point.

Minor corrections:

- Remove collagen in subscript "bone-collagen" throughout the text
- figure 6 should read $\delta^{66}\text{Zn}$, not $\delta^{66}\text{Z}$
- remove regression line fig 4

This is a nice work, Vincent Balter

Reviewer #2 (Remarks to the Author):

The work by Leichter et al is a proxy validation study of $\delta^{15}\text{N}$ measurements of enamel organics, as well as an application of the $\delta^{15}\text{N}$ to a fossil setting. The work builds on previous methodological studies (particularly Leichter et al 2021, Chem. Geol., and others), which show, intriguingly, that enamel $\delta^{15}\text{N}$ is measurable by the oxidation-denitrification method, and other studies which show that enamel is relatively resistant to diagenesis. Care is taken with sample selection to prevent confounding effects such as weaning, etc., and the case is made, convincingly on modern material, for $\delta^{15}\text{N}$ -patterning which reflects the expected trends in trophic level, amongst several other potentially confounding effects. On their own, the modern data would warrant publication in Nat Comms, and the modern data shown in Fig 2 are particularly nice.

In general, this method presents a conceptual leap in deep-time faunal isotope studies.

I have a couple of minor comments and suggestions, below. However, my one main concern is considering the paleo-data, given Figure 2.

1) If the authors wish to maintain their claim that the method is widely applicable to fossil contexts, then they may wish to either obtain more data from fossil contexts (currently $n=4$ is too little); and/or perhaps be more slightly careful to qualify these statements - where they appear in the MS. For instance, I performed a Pearson's test for significance of correlation, and also Kendall's rank correlation tau, and the relationship is not significant. Please can the authors check this, and comment on it?

2) I can accept that regression residuals of the fossil fauna might be in range of the modern mammals (line 274), but those four THM fossil datapoints seem to have higher scatter, and are at the extreme of the plot in Figure 2. Why? This needs to be discussed, and more data are necessary to infer that the method is as generally applicable to fossil contexts as is perhaps claimed here. Hence, the implication that the application to "Fossil ecosystems" (Abstract, line 48, also the title) is universal might need to be tempered.

Lines 61-73. It is a minor concern, but this opening paragraph is a little vague, and goes straight

into issues with Ca and Zn isotope proxies of trophic level. Perhaps it might be better to start with the second paragraph (74 onwards), which establishes quite nicely the general importance of $\delta^{15}\text{N}$ as trophic level proxy, before going into the specific reasons why $\delta^{15}\text{N}$ in enamel is important (one of which is the issues with Ca and Zn). Furthermore, the major reason why this new method is so important, is that the poor preservation of (bone/dentine) collagen in deep time, and relatively small amounts of N in tooth enamel, have previously precluded $\delta^{15}\text{N}$ being used as a viable proxy. I think that this should justifiably be the leading angle - not necessarily Zn or Ca.

826. Friedli et al is now a very old reference for $\delta^{13}\text{CO}_2$ shifts, which is now considerably more negative than 1980s values. There is more recent ice core compilation data, e.g. in Hare et al (2018) Nat Comms.

Supplementary 86: "OF" correction of capitalisation

Thank you, and good luck!

Reviewer #3 (Remarks to the Author):

This is a nice, interesting paper, that increases the geochemical tools available to understand trophic structure of past ecosystems. Reconstruction of trophic networks is usually limited to protein (mainly collagen, but also keratin) preservation, which, in the best of cases, extends back only until the mid-Pliocene (although good quality collagen of pre-Pleistocene samples are rare). Having, therefore, a method to assess trophic information without the need for collagen preservation is novel and exciting. By providing paired data for both d^{15}N of enamel organic matter and d^{15}N of collagen of modern mammals, this paper provides valuable data showing that both signals are basically undistinguishable.

I commend the authors for their clear writing and cautious approach and so I only have a few questions and minor suggestions. My only criticism is that the authors remain in the rather "report-type" of paper, instead of proposing more thorough explanations for some of the interesting results they got. For instance, I would have liked to see a deeper discussion on why the "mixed-feeders" have higher $\text{d}^{15}\text{N}_{\text{enamel}}$ values than browsers and grazers? Some of the mixed-feeders have d^{15}N values as high (or even higher) as those observed in carnivores. Does the difference stem from the fact that they are not strictly folivores (if one can include grass within the "folivore" category)? Why is the $\text{d}^{15}\text{N}_{\text{enamel}}$ values of the THM browsers higher than that of the grazers from the same locality? This is interesting because it differs from what they obtained in modern African mammals. I also find it interesting that the range of $\text{d}^{15}\text{N}_{\text{enamel}}$ values of baboons is narrower than that of mixed feeders, in spite of the large range of items (including even elements of animal origin) in baboons' diets.

These omissions, which are easily incorporated however, do not diminish the value of this paper. Indeed, it opens the door for exploration of this technique in other vertebrate systems and I think it will be of interest to others in the paleoecology community and beyond. The authors claim that enamel d^{15}N is as strong a trophic level proxy as collagen d^{15}N , and they provide enough data supporting that assertion. It is remarkable that the overall differences among trophic categories are observed even considering specimens from different localities (with surely different d^{15}N baseline values). Collagen d^{15}N and now $\text{d}^{15}\text{N}_{\text{enamel}}$ are limited, of course, in the resolution they can achieve, and in particular, omnivory and mixed-feeding behaviors are in the blind spot of the technique. Thus, their assessment that $\text{d}^{15}\text{N}_{\text{enamel}}$ will clarify animal-resource consumption by Australopithecus is perhaps a bit optimistic (but they might prove me wrong). The $\text{d}^{15}\text{N}_{\text{enamel}}$ data of modern mammals clearly differentiates folivores (browsers and grazers) from carnivores, but less so carnivores from mixed-feeding herbivores (unless at the lower and higher extreme values of those feeding categories). Some elephants and springboks, for instance, have $\text{d}^{15}\text{N}_{\text{enamel}}$ as high as lions (Table 1). This is just to say that as wonderful as this proxy is, perhaps a short mention of its limitations would be worth to be included at the end of the discussion (or conclusions).

Attached, I provide some comments, questions, and suggestions that I hope the authors will find useful. I also would like to draw the attention to the existence of d15N data from modern Amazonian mammals (Tejada et al 2020, PNAS). Although that paper focuses on d13C, there are keratin d15N data that might be relevant to this paper (particularly, in the discussion of mixed-feeders having higher d15N values than browsers and grazers).

Two couple of details that need clarification include:

- Why was the carbonate d13C untreated? Why they did not get rid of the organics and secondary carbonates?
- Were the d13C data corrected for the Suess effect? For the modern specimens, depending on when these were collected, the d13C values can change significantly. Please report both the raw and the corrected data.

More in the attached file. Good luck!

Point by Point Response to Reviewers' comments for the manuscript “*Tooth enamel nitrogen isotope composition records trophic position: a tool for reconstructing food webs*”:

Reviewer #1 (Remarks to the Author):

Review of the manuscript “Nitrogen isotopic composition of tooth enamel organic matter records trophic position in modern and fossil ecosystems” by Leichter et al. for *Communications Biology*.

This is an interesting paper that pushes analytical limits of measuring N isotopes in modern and fossil enamel. I have no major concerns and would recommend publication providing minor revisions. The main point is that I found the first paragraph of the introduction a bit clumsy. Regarding Ca and Zn isotopes, the authors write that “data documenting baseline variation ... is limited. As such, a reliable, well-characterized geochemical proxy for determining trophic position in deep time remains elusive.” This is clearly not the case for Ca isotopes which are particularly efficient for very old trophic systems. Thomas Tütken must know about this as he co-authored one important paper on Ca isotopes in dinosaurs. The reason why Ca isotopes are very efficient at reconstructing trophic relationships is when bone comes with the whole prey, which is not necessarily the case regarding mammals. The authors should acknowledge this point.

We have addressed this concern by reorganizing the order of the paragraphs (as suggested by reviewer 2) so that the first paragraph of the Introduction focuses on nitrogen only (Lines 60-72). The discussion of the use of “non-traditional” stable isotopes (i.e., Ca and Zn) for trophic level reconstruction has been moved to the second paragraph and the text of this paragraph has been modified. Additionally, we clarify that Ca (and increasingly Zn) are effective and useful tools for reconstructing trophic relationships in the past (as demonstrated by e.g., Balter et al., 2018, Martin et al., 2015, 2018; Hassler et al., 2018) and that they have added significantly to our geochemical toolkit. We have removed the sentence stating that no robust proxy exists for reconstructing trophic level in deep time, following the suggestion of the reviewer (Lines 77-81). Nevertheless, although the number of studies documenting baseline variation in Ca and Zn isotopes continues to expand, N isotopes have been widely used across a variety of disciplines for decades, permitting us to better interpret our data as well as understand its limitations.

Minor corrections:

- Remove collagen in subscript “bone-collagen” throughout the text

We chose to specify “bone-collagen” in subscript throughout the text to differentiate it from “dentin-collagen” which was analyzed for the Tam Hay Marklot fossil specimens. Therefore, we have retained this subscript throughout to avoid any ambiguity.

- figure 6 should read $d_{66}Zn$, not $d_{66}Z$

This has been corrected in Fig. 6.

- remove regression line fig 4

This regression line itself has been removed from Fig. 4 and information about sample size and p -value have been added.

This is a nice work, Vincent Balter

Thank you very much for this helpful review.

Reviewer #2 (Remarks to the Author):

The work by Leichliter et al. is a proxy validation study of $\delta^{15}\text{N}$ measurements of enamel organics, as well as an application of the $\delta^{15}\text{N}$ to a fossil setting. The work builds on previous methodological studies (particularly Leichliter et al 2021, Chem. Geol., and others), which show, intriguingly, that enamel $\delta^{15}\text{N}$ is measurable by the oxidation-denitrification method, and other studies which show that enamel is relatively resistant to diagenesis. Care is taken with sample selection to prevent confounding effects such as weaning, etc., and the case is made, convincingly on modern material, for $\delta^{15}\text{N}$ -patterning which reflects the expected trends in trophic level, amongst several other potentially confounding effects. On their own, the modern data would warrant publication in Nat Comms, and the modern data shown in Fig 2 are particularly nice.

In general, this method presents a conceptual leap in deep-time faunal isotope studies.

I have a couple of minor comments and suggestions, below. However, my one main concern is considering the paleo-data, given Figure 2.

1) If the authors wish to maintain their claim that the method is widely applicable to fossil contexts, then they may wish to either obtain more data from fossil contexts (currently $n=4$ is too little); and/or perhaps be more slightly careful to qualify these statements - where they appear in the MS. For instance, I performed a Pearson's test for significance of correlation, and also Kendall's rank correlation tau, and the relationship is not significant. Please can the authors check this, and comment on it?

Thank you for this helpful assessment of the paleo-dataset. Regarding the reviewer's comments on relationship between $\delta^{15}\text{N}_{\text{enamel}}$ and $\delta^{15}\text{N}_{\text{collagen}}$ values in the fossil specimens, we have added a statement that the correlation between the two is not-significant and report the results of the relevant statistical tests in the Results section (Lines 270-272). These results are addressed briefly in the discussion section of the manuscript (Lines 502-504). We suspect that perhaps poor preservation of dental collagen in some of the fossil specimens may have contributed to this result (of 23 attempted extractions only 4 specimens yielded sufficient material for isotopic measurement), as well as to the scatter in the fossil data $\delta^{15}\text{N}$ data compared to that of the modern mammals (see our response to the reviewer's second comment for additional details). As the reviewer notes, the small sample size $n = 4$ (samples paired $\delta^{15}\text{N}_{\text{enamel}}$ and $\delta^{15}\text{N}_{\text{bone-collagen}}$ values) makes the relationship difficult to evaluate in the fossil dataset. Per the reviewer's suggestion we focus the manuscript on ground-truthing the oxidation-denitrification method using modern tooth enamel and report preliminary results of fossil data to demonstrate to the reader the potential of the method. We have tempered our statements regarding the application of this method to fossil contexts given the small size of our paleo-dataset. We have given the manuscript a new title, "*Tooth enamel nitrogen isotope composition records trophic position: a tool for reconstructing food webs*", we have revised the Abstract, and we have modified the text where appropriate throughout the Results and Discussion sections to qualify our statements about the paleo-data and address concerns about the small sample size. These changes emphasize that this study is focused on establishing tooth enamel nitrogen isotope composition as a new tool for reconstructing trophic level and present the paleontological data as a preliminary test for application to the fossil record.

While we generally agree with the reviewer's assessment that this is a too small of a dataset to draw overly broad conclusions about universal applicability of $\delta^{15}\text{N}_{\text{enamel}}$ this proxy at this time, we nevertheless feel that the trophic enrichment between herbivores and carnivore evident in the THM fossil $\delta^{15}\text{N}_{\text{enamel}}$ dataset ($n = 10$) shows promise for the application of this method to fossil record. Moreover, we specifically chose to use the THM fossil specimens because i) Bourgon et al. (2020) had already demonstrated that enamel preservation in the specimens was very good (despite poor organic preservation of bone and dentin), ii) because Zn isotope values had been measured in all 10 of the samples we analyzed, and iii) because $\delta^{15}\text{N}_{\text{bone-collagen}}$ values existed for some of the specimens. This permitted us to reconstruct trophic level using both isotopic proxies ($\delta^{15}\text{N}_{\text{enamel}}$ and $\delta^{66}\text{Zn}_{\text{enamel}}$) as

well as to evaluate the relationship between them. Such multi-isotopic studies represent, in our opinion an important future direction for paleo-dietary studies.

2) I can accept that regression residuals of the fossil fauna might be in range of the modern mammals (line 274), but those four THM fossil datapoints seem to have higher scatter, and are at the extreme of the plot in Figure 2. Why? This needs to be discussed, and more data are necessary to infer that the method is as generally applicable to fossil contexts as is perhaps claimed here. Hence, the implication that the application to “Fossil ecosystems” (Abstract, line 48, also the title) is universal might need to be tempered.

We suspect that the high scatter in the fossil data may be related, at least in part, to poor preservation of dental collagen. For example, Bourgon et al. 2020 remark that the high $\delta^{15}\text{N}_{\text{collagen}}$ value of 10.6 ‰ for the Muntjac (which would reflect a trophic level higher than that of the other three measured individuals) was “surprising and unexpected” (Bourgon et al. 2020; Supplementary Information, pg. 17). In contrast, we measured a lower $\delta^{15}\text{N}_{\text{enamel}}$ value of 7.6 ‰ for the same individual, which falls within the range of values more typically observed for modern herbivores and within the range of the other THM herbivores.

Additionally, N content for all fossil specimens was similar to that of modern tooth enamel, indicating good preservation of enamel organic matter. Again, as the reviewer points out, the fossil dataset is too small to draw broad conclusions and we have reworded both the Abstract and the Title to better represent the data.

Lines 61-73. It is a minor concern, but this opening paragraph is a little vague, and goes straight into issues with Ca and Zn isotope proxies of trophic level. Perhaps it might be better to start with the second paragraph (74 onwards), which establishes quite nicely the general importance of $\delta^{15}\text{N}$ as trophic level proxy, before going into the specific reasons why $\delta^{15}\text{N}$ in enamel is important (one of which is the issues with Ca and Zn). Furthermore, the major reason why this new method is so important, is that the poor preservation of (bone/dentine) collagen in deep time, and relatively small amounts of N in tooth enamel, have previously precluded $\delta^{15}\text{N}$ being used as a viable proxy. I think that this should justifiably be the leading angle - not necessarily Zn or Ca.

Thank you for the helpful suggestions. Reviewer 1 had a similar comment and we have addressed this concern by reorganizing the order of the paragraphs so that the first paragraph of the Introduction focuses on nitrogen only. The brief discussion of Ca and Zn has been moved to the second paragraph and the text of this paragraph has been modified (Lines 77-81).

826. Friedli et al is now a very old reference for $\delta^{13}\text{C}_{\text{CO}_2}$ shifts, which is now considerably more negative than 1980s values. There is more recent ice core compilation data, e.g. in Hare et al (2018) Nat Comms.

We have updated the reference for $\delta^{13}\text{C}_{\text{atm}}$ to Hare et al. 2018 per the reviewer’s suggestion. However, we would like to note that the reported age of 38.4–13.4 ka for the THM fossil specimens is not very precise which adds considerable uncertainty about the appropriate correction for $\delta^{13}\text{C}_{\text{CO}_2}$ change over this period of time. This age range spans 25 ka and several periods of major climatic change resulting in significant variation (as much as 1.0 ‰) in atmospheric $\delta^{13}\text{C}$ (Hare et al. 2018). Thus, while we have corrected the fossil values by 2 ‰ to account for recent, anthropogenically induced shifts in atmospheric $\delta^{13}\text{C}$ (Suess effect), in the absence of more precise dates for the fossil specimens this adjustment must be considered approximate at best because it is unclear what the $\delta^{13}\text{C}_{\text{CO}_2}$ value to use for this time range.

Supplementary 86: “OF” correction of capitalisation

This typo has been corrected.

Thank you, and good luck!

Thank you very much for your thoughtful comments which will greatly improve this manuscript.

Reviewer #3 (Remarks to the Author):

This is a nice, interesting paper, that increases the geochemical tools available to understand trophic structure of past ecosystems. Reconstruction of trophic networks is usually limited to protein (mainly collagen, but also keratin) preservation, which, in the best of cases, extends back only until the mid-Pliocene (although good quality collagen of pre-Pleistocene samples are rare). Having, therefore, a method to assess trophic information without the need for collagen preservation is novel and exciting. By providing paired data for both $\delta^{15}\text{N}$ of enamel organic matter and $\delta^{15}\text{N}$ of collagen of modern mammals, this paper provides valuable data showing that both signals are basically undistinguishable.

Comment #1: I commend the authors for their clear writing and cautious approach and so I only have a few questions and minor suggestions. My only criticism is that the authors remain in the rather “report-type” of paper, instead of proposing more thorough explanations for some of the interesting results they got. For instance, I would have liked to see a deeper discussion on why the “mixed-feeders” have higher $\delta^{15}\text{N}_{\text{enamel}}$ values than browsers and grazers? Some of the mixed-feeders have $\delta^{15}\text{N}$ values as high (or even higher) as those observed in carnivores. Does the difference stem from the fact that they are not strictly folivores (if one can include grass within the “folivore” category)? Why is the $\delta^{15}\text{N}_{\text{enamel}}$ values of the THM browsers higher than that of the grazers from the same locality? This is interesting because it differs from what they obtained in modern African mammals. I also find it interesting that the range of $\delta^{15}\text{N}_{\text{enamel}}$ values of baboons is narrower than that of mixed feeders, in spite of the large range of items (including even elements of animal origin) in baboons’ diets.

An earlier draft of the manuscript included a more detailed discussion of the interesting variation in $\delta^{15}\text{N}_{\text{enamel}}$ values amongst the herbivores in the main text; however, we received feedback from co-authors that its inclusion disrupted the flow and main argument of the manuscript, which is focused on establishing the validity of the proxy rather than explicitly discussing the ecology of the analyzed taxa, and also made the manuscript overly long. However, we find the reviewers questions both interesting and valid. In order to address them, we have added text to the main manuscript (Lines 380-387 and 407-411) discussing this topic and also directing the reader to the Supplementary materials for an expanded discussion of this topic. We discuss the $\delta^{15}\text{N}_{\text{enamel}}$ values of the THM fossil browsers in the discussion of the main text (Lines 515-520), but we remain cautious with our interpretations of these data given the small fossil dataset.

We have also modified our discussion of baboon $\delta^{15}\text{N}_{\text{enamel}}$ values and note that the narrow (and relatively low) range of $\delta^{15}\text{N}_{\text{enamel}}$ values that we observed in the baboons is driven by infrequent intake of and limited access to animal resources during the time of tooth formation (see also Lüdecke & Leichliter et al., 2022), as well as our limited sample size and the fact that the baboons only derive from two localities. Although a moderate consumption of animal resources can impact $\delta^{15}\text{N}_{\text{enamel}}$ values, as demonstrated by Leichliter et al. (2021) in a feeding experiment using rodents (animals that received meat and insect-based pellets were enriched in $\delta^{15}\text{N}$ over those that received plant-based pellets despite the fact that the meat and insect pellets contained only 25% animal product), this intake needs to be relatively consistent throughout tissue formation. Infrequently consumed foods are unlikely to impact bulk $\delta^{15}\text{N}$ values in and, despite significant dietary flexibility, baboons nevertheless consume mostly plant material (Hill and Dunbar, 2002), especially as young adults of low rank.

Full Citations for this response:

Hill, R. A., & Dunbar, R. I. M. (2002). Climatic determinants of diet and foraging behaviour in baboons. *Evolutionary Ecology*, 16(6), 579–593. <https://doi.org/10.1023/A:1021625003597>

Leichliter, J. N., Lüdecke, T., Foreman, A. D., Duprey, N. N., Winkler, D. E., Kast, E. R., Vonhof, H., Sigman, D. M., Haug, G. H., Clauss, M., Tütken, T., & Martínez-García, A. (2021). Nitrogen isotopes in tooth enamel record diet and trophic level enrichment: Results from a controlled feeding experiment. *Chemical Geology*, 563, 120047. <https://doi.org/10.1016/j.chemgeo.2020.120047>

Lüdecke, T., Leichliter, J. N., Aldeias, V., Bamford, M. K., Biro, D., Braun, D. R., Capelli, C., Cybulski, J. D., Duprey, N. N., Ferreira da Silva, M. J., Foreman, A. D., Habermann, J. M., Haug, G. H., Martínez, F. I., Mathe, J., Mulch, A., Sigman, D. M., Vonhof, H., Bobe, R., ... Martínez-García, A. (2022). Carbon, nitrogen, and oxygen stable isotopes in modern tooth enamel: A case study from Gorongosa National Park, central Mozambique. *Frontiers in Ecology and Evolution*, 10, 1107. <https://doi.org/10.3389/FEVO.2022.958032>

Comment #2: These omissions, which are easily incorporated however, do not diminish the value of this paper. Indeed, it opens the door for exploration of this technique in other vertebrate systems and I think it will be of interest to others in the paleoecology community and beyond. The authors claim that enamel $\delta^{15}\text{N}$ is as strong a trophic level proxy as collagen $\delta^{15}\text{N}$, and they provide enough data supporting that assertion. It is remarkable that the overall differences among trophic categories are observed even considering specimens from different localities (with surely different $\delta^{15}\text{N}$ baseline values). Collagen $\delta^{15}\text{N}$ and now $\delta^{15}\text{N}_{\text{enamel}}$ are limited, of course, in the resolution they can achieve, and in particular, omnivory and mixed-feeding behaviors are in the blind spot of the technique. Thus, their assessment that $\delta^{15}\text{N}_{\text{enamel}}$ will clarify animal-resource consumption by *Australopithecus* is perhaps a bit optimistic (but they might prove me wrong). The $\delta^{15}\text{N}_{\text{enamel}}$ data of modern mammals clearly differentiates folivores (browsers and grazers) from carnivores, but less so carnivores from mixed-feeding herbivores (unless at the lower and higher extreme values of those feeding categories). Some elephants and springboks, for instance, have $\delta^{15}\text{N}_{\text{enamel}}$ as high as lions (Table 1). This is just to say that as wonderful as this proxy is, perhaps a short mention of its limitations would be worth to be included at the end of the discussion (or conclusions).

We have added text to the discussion section (Lines 531-537), acknowledging the shortcomings of $\delta^{15}\text{N}$ analysis of bulk tissue in resolving the diets of fossil taxa with omnivorous or mixed feeding behaviors. We further suggest that the best way forward is a multi-isotopic approach for dietary reconstruction of generalist feeders, such as combining bulk $\delta^{15}\text{N}$ and Zn (or other) isotopic systems.

Comment #3: I also would like to draw the attention to the existence of $\delta^{15}\text{N}$ data from modern Amazonian mammals (Tejada et al 2020, PNAS). Although that paper focuses on $\delta^{13}\text{C}$, there are keratin $\delta^{15}\text{N}$ data that might be relevant to this paper (particularly, in the discussion of mixed-feeders having higher $\delta^{15}\text{N}$ values than browsers and grazers).

Thank you for the suggestions, we have added reference to Tejada et al. 2020 throughout the manuscript, specifically with reference to the discussion of mixed feeders.

Comment #4: Why was the carbonate $\delta^{13}\text{C}$ untreated? Why they did not get rid of the organics and secondary carbonates?

We compared our untreated fossil $\delta^{13}\text{C}_{\text{enamel}}$ values for all samples ($n = 10$) to those published by Bourgon et al. 2020 for the same specimens (although not exactly the same aliquot of tooth enamel powder), that were pretreated for the removal of organics and secondary carbonates. This data was not included in the originally submitted version of the manuscript but is now presented in the Supplementary Materials (Fig. S5 and Lines 196-200). Treated and untreated sample $\delta^{13}\text{C}_{\text{enamel}}$ values show a significant, strong positive correlation ($R^2 = 0.99$, $p < 0.001$), thus we feel confident in our $\delta^{13}\text{C}_{\text{enamel}}$ values for fossil specimens.

Additionally, it has become clear from experiments (Snoeck and Pillegrini, 2015; Pellegrini and Snoeck, 2016), that the typical NaOCl treatment step actually adds carbon that was not present before (the effect is low for enamel, but severe for bone). As it is meant to eliminate organics, which are minimal even in modern enamel, thus this cleaning step is sometimes redundant.

Moreover, for this study, we wanted to measure $\delta^{13}\text{C}_{\text{enamel}}$ in the exact same aliquot of tooth enamel measured for $\delta^{15}\text{N}_{\text{enamel}}$. Since we require 5mg of material per $\delta^{15}\text{N}_{\text{enamel}}$ measurement we wanted to avoid sample loss as much as possible (which can be significant during the pre-treatment for the removal of secondary carbonates) in order to ensure enough material for the measurement of $\delta^{15}\text{N}_{\text{enamel}}$ duplicates and avoid any treatment that might somehow alter the endogenous organic matter and thereby effect the $\delta^{15}\text{N}_{\text{enamel}}$ values of the samples.

Comment #5: Were the $\delta^{13}\text{C}$ data corrected for the Suess effect? For the modern specimens, depending on when these were collected, the $\delta^{13}\text{C}$ values can change significantly. Please report both the raw and the corrected data.

Since the exact year of collection for all of the modern African mammals was not known, enamel and collagen $\delta^{13}\text{C}$ values themselves were not corrected for the Suess effect. Instead, we corrected the C_3 and C_4 end-members (green bars at the top of Fig. 3) based on bioapatite $\delta^{13}\text{C}$ data from Cerling et al. 2003 for African bovids (specimens collected between 1997 to 2000) and an atmospheric $\delta^{13}\text{C}_{\text{CO}_2}$ of -7.1 ‰ for the period between 1950 and 1970 (Dombrosky et al., 2019), which is when the mammals used in this study were probably collected according to museum records.

Similarly, in Fig. 5, fossil enamel and collagen $\delta^{13}\text{C}$ values themselves have not been corrected, instead only the C_3 and C_4 end-members (green bars at the top of Fig. 5) have been adjusted to reflect the fossil-fuel-induced shift in the $\delta^{13}\text{C}$ of atmospheric CO_2 of ~ -2.0 ‰ compared to pre-industrial times.

The only instance in which modern and fossil $\delta^{13}\text{C}$ values were directly compared is in the Supplementary Information in Fig. S5. In this case, we chose to adjust the $\delta^{13}\text{C}_{\text{enamel}}$ values of the four THM fossils directly. A correction of -0.7 ‰ was applied to permit direct comparison of the fossil specimens and the African mammals, which were collected between 1950 and 1970. However, we would like to note that the reported age of 38.4–13.4 ka for the THM fossil specimens is not very precise. This age range spans 25 ka and several periods of major climatic change resulting in significant variation (as much as 1.0 ‰) in atmospheric $\delta^{13}\text{C}$ (Hare et al. 2018). Thus, while we have corrected the fossil values for recent, anthropogenically induced shift in the $\delta^{13}\text{C}$ of atmospheric CO_2 , in the absence of more precise dates for the fossil specimens this adjustment must be considered approximate at best.

More in the attached file. Good luck!

- Julia Tejada

Thank you so much for your very detailed comments which will greatly improve this manuscript.

Response to Reviewer 3 comments in attached PDF of manuscript:

Lines 71-73: there is an omission here of what it is arguably the most reliable method to reconstruct trophic positions: AACCSIA. Include this if space allows

We agree that AA-CSIA is one of the most reliable methods for reconstructing trophic position, however, it is only applicable to fossil specimens with good collagen preservation (i.e., typically

< 100 ka). Since we are specifically referring to methods applicable in “deep time” we did not mention AA-CSIA in our list of “deep time” proxies in our originally submitted manuscript. We have re-written the first two introductory paragraphs and now mention amino acid specific analyses (AA-CSIA) in the first paragraph (Lines 68, 90-91).

Lines 117 – 120: citation needed

The following citations have been added here:

Sealy, J. C., van der Merwe, N. J., Thorp, J. A. L. & Lanham, J. L. Nitrogen isotopic ecology in southern Africa: Implications for environmental and dietary tracing. *Geochimica et Cosmochimica Acta* 51, 2707-2717, doi:10.1016/0016-7037(87)90151-7 (1987).

Ambrose, S. H. Effects of diet, climate and physiology on nitrogen isotope abundances in terrestrial foodwebs. *Journal of Archaeological Science* 18, 293-317, doi:10.1016/0305-4403(91)90067-Y (1991).

Ambrose, S. H. & DeNiro, M. J. The isotopic ecology of East African mammals. *Oecologia* 69, 395-406, doi:10.1007/BF00377062 (1986).

Heaton, T. H. E., Vogel, J. C., Von La Chevallerie, G. & Collett, G. Climatic influence on the isotopic composition of bone nitrogen. *Nature* 322, 822-823, doi:10.1038/322822a0 (1986).

Ambrose, S. H. & DeNiro, M. J. Reconstruction of African human diet using bone collagen carbon and nitrogen isotope ratios. *Nature* 319, 321-324, doi:10.1038/319321a0 (1986).

Schwarcz, H. P., Dupras, T. L. & Fairgrieve, S. I. ¹⁵N Enrichment in the Sahara: In Search of a Global Relationship. *Journal of Archaeological Science* 26, 629-636, doi:10.1006/JASC.1998.0380 (1999).

Balter, V., Laurent, A. E., Aehéié Ne Fouillet, S. & Le'cuyer, C. L. c. Box-modeling of ¹⁵N/¹⁴N in mammals. *Oecologia* 147, 212-222, doi:10.1007/s00442-005-0263-5 (2006).

Lines 124 – 131: specify that with "mixed feeders" they refer to herbivores feeding on both C3 and C4 plants. Or do they mean feeding on different plant organs? Not clear

We now specify in the text (Lines 126-128) that we use the term browsers, grazers, and mixed feeders for herbivores that consume mainly C₃-plants, mainly C₄-plants, and a mixture of both C₃- and C₄-plants, respectively.

Fig. 1: add n= for each feeding group category.

Sample size (*n*) has been added for each feeding group for Fig. 1, as well as in Fig. 2, Fig. 4, Fig. 5, and Fig. 6).

Fig. 1: why is this elephant's collagen value so high?

As discussed in the main text of the manuscript (Lines 405-407) elephant $\delta^{15}\text{N}$ values are known to vary quite significantly. The elephant with the high $\delta^{15}\text{N}_{\text{bone-collagen}}$ value (12.3 ‰) to which the reviewer refers also has quite a high $\delta^{15}\text{N}_{\text{enamel}}$ value (10 ‰). Thus, whatever is driving the high $\delta^{15}\text{N}_{\text{bone-collagen}}$ value is also reflected in the enamel organic matter.

Lines 206-208 (Fig. 1 caption): not quite, I would eliminate this last sentence

We do not argue that the pattern of ^{15}N enrichment is identical between collagen and enamel across dietary groups, we only argue that they are similar which is accurate here. Carnivores are most enriched in ^{15}N , followed by mixed feeding, grazing, and then browsing herbivores. This relative pattern is consistent across both collagen and enamel. We feel that this point is important in demonstrating that the enamel $\delta^{15}\text{N}$ records similar information to bulk collagen analysis, therefore we chose to retain the last sentence of the figure caption.

Fig. 2: add p-value for the correlation

We have added the p -value to Fig. 2.

Lines 216 – 217 (Fig. 2 caption): delete last sentence

This sentence has been deleted.

Lines 234 – 235: is this discussed later on?

This (i.e., the isotopic spacing in $\delta^{13}\text{C}$ between enamel and collagen in carnivores versus herbivores) is discussed in more detail in the Supplementary materials under the section “Variation in $\delta^{13}\text{C}$ values in tooth enamel”. The observed offset between the herbivores and carnivores is related to differences in the composition of the tissue being measured and the diets of these two groups. It has been well-established that the carbon in the inorganic (carbonate) component of enamel reflects whole diet, whereas carbon in collagen reflects only the protein component of the diet. Thus, carnivores, who consume a protein rich diet have more similar enamel and collagen values compared to herbivores, who consume a relatively protein-poor (but carbohydrate rich) plant-based diet. This is not discussed in detail in the main text as it is not especially germane to our primary arguments.

Fig. 3: Symbols as in Fig. 1?

A legend with the symbols (same as in Fig. 1) have been added to Fig. 3

Fig. 5: add $n=$ for each taxon here

Sample size ($n =$) has been added for each diet group in Fig. 5.

Lines 303 – 304 (Fig. 5 caption): mention $\delta^{13}\text{C}$ Suess correction in the text. Was it done for all the $\delta^{13}\text{C}$ data? what about the yellow symbol?

Please see details regarding the Suess correction in our response to Comment #5.

We have added more specific information in the caption to clarify which color indicates which feeding category for the herbivores. The “yellow” symbol is a mixed feeder.

Fig. 6: include p value for regression

p -value has been added for $\delta^{15}\text{N}_{\text{enamel}}$ versus $\delta^{66}\text{Zn}_{\text{enamel}}$ regression in Fig. 6.

Lines 340 – 343: well, in African system perhaps. See Tejada et al 2020 (PNAS) on $\delta^{15}\text{N}$ of Amazonian mammals

We have changed the text (Line 345) to specify that bone collagen $\delta^{15}\text{N}$ is well understood in “African ecosystems”.

Lines 377 – 381: fat content of fruits. There is relevant $\delta^{15}\text{N}$ data from modern Amazonian mammals that the authors can compare with (Tejada et al. 2020)

We have added “macronutrient composition of diet” (Lines 382-383) to the set of examples given for dietary factors driving variation in the $\delta^{15}\text{N}$ of herbivore tissues, which should encompass e.g., the fat content of fruits and have also added Tejada et al. 2020 as a citation. Additionally, we have added “fat-content” to the list of examples given for variation in plant nutritional quality (Lines 383-387).

Lines 400 – 403: yes it does, again, see data of Amazon mammals

We now state that mixed feeders have higher $\delta^{15}\text{N}$ values than other herbivores in both Africa and South America (Lines 411-414) and cite Tejada et al. 2020. We further specify that “our $\delta^{15}\text{N}_{\text{enamel}}$ dataset for one given locality in Africa is too small to test if this holds true *for tooth enamel* amongst mixed feeders more broadly”.

Lines 423 – 424: mixed feeders in particular

This has now been noted in the text (Lines 391-392)

Lines 430 – 432: I don't quite follow this explanation, baboons samples come from only 2 localities and the range of $\delta^{15}\text{N}_{\text{enamel}}$ is not large

We agree and we have removed the statement that “the specimens come from different sampling localities across Africa” as it is not relevant in this case.

Lines 442 – 444: Or not. Perhaps it has to do with the fact that hyenas eat bones.

We considered the possibility that bone consumption might have contributed to the elevated $\delta^{15}\text{N}_{\text{enamel}}$ values that we observed in the spotted hyenas. The primary source of nitrogen in bones should come from either bone collagen (which was measured) or bone marrow. Unless considerable fractionation occurs during the metabolism of collagen in hyena digestive system, it is unlikely that this is driving the higher $\delta^{15}\text{N}_{\text{enamel}}$ values. We found very little data on bone marrow $\delta^{15}\text{N}$ values in the literature, but the values that we did find are not especially elevated in $\delta^{15}\text{N}$ (Drucker and Bocherens, 2004). Thus, while we cannot rule out bone consumption as a contributing factor, we did not find strong evidence that this caused the elevated $\delta^{15}\text{N}_{\text{enamel}}$ values. Wißing et al. (2019) found that Pleistocene cave hyenas have higher $\delta^{15}\text{N}_{\text{collagen}}$ values than coexisting carnivore species, but this enrichment has not been documented in modern spotted hyenas and lions in Africa (Codron et al., 2016), and it is known that both predator species exhibit a large overlap in prey choice (Hayward et al., 2006). So, although the observed differences are potentially related to dietary differences (such as higher consumption of bone in hyenas), we think the more likely explanation is the early the tooth mineralization (11 to 13 months) relative to tooth eruption schedule (between 12 and 14 months) in spotted hyenas resulting in a nursing signal in our $\delta^{15}\text{N}_{\text{enamel}}$ values. A discussion of this is now included in the Supplementary Materials (Lines 113-128).

Full citations for this response:

Codron, D., Codron, J., Sponheimer, M., & Clauss, M. (2016). Within-population isotopic niche variability in savanna mammals: Disparity between carnivores and herbivores. *Frontiers in Ecology and Evolution*, 4 (FEB). <https://doi.org/10.3389/fevo.2016.00015>

Drucker, D., & Bocherens, H. (2004). Carbon and Nitrogen Stable Isotopes as Tracers of Change in Diet Breadth during Middle and Upper Palaeolithic in Europe. *International Journal of Osteoarchaeology International. Journal of. Osteoarchaeol*, 14, 162–177. <https://doi.org/10.1002/oa.753>

Hayward, M. W. (2006). Prey preferences of the spotted hyaena (*Crocuta crocuta*) and degree of dietary overlap with the lion (*Panthera leo*). *Journal of Zoology*, 270(4), 606–614. <https://doi.org/10.1111/J.1469-7998.2006.00183.X>

Wißing, C., Rougier, H., Crevecoeur, I., Germonpré, M., Naito, Y. I., Semal, P., & Bocherens, H. (2016). Isotopic evidence for dietary ecology of late Neandertals in North-Western Europe. *Quaternary International*, 411, 327–345. <https://doi.org/10.1016/J.QUAINT.2015.09.091>

Lines 503 – 505: the range of $\delta^{15}\text{N}_{\text{enamel}}$ values for omnivores encompass that of herbivores. Indeed, there are not significant differences in $\delta^{15}\text{N}_{\text{enamel}}$ between herbivores and omnivores

This information and statistical significance is now reported in the Results section (Lines 281-283).

Lines 509 – 510: It is only higher than the values for grazers (but not for the mixed feeders)

We now specify this in the text (Lines 511-513).

Lines 513 – 516: maybe, but with an $n=2$ it is hard to discern any cogent pattern. In any case, this is beyond the resolution possible to achieve with any bulk $\delta^{15}\text{N}$ analyses (enamel, collagen, etc.)

We agree with the reviewer and have noted that any hypotheses about the THM fossil browser $\delta^{15}\text{N}_{\text{enamel}}$ values should be considered cautiously as more data is required

.

Lines 518 – 522: based on morphology? Not clear

We now state “based on taxonomy and dental morphology” (Lines 522-523).

Lines 617: justify why was the sample untreated? why was the organic material (and secondary carbonates) not removed?

Please refer to our response to **Comment #4**

Table 1: Put $\delta^{13}\text{C}$ columns (and $\delta^{15}\text{N}$ columns) next to each other. It makes it easier to read and to compare

Table 1 and 2 have been rearranged accordingly.

REVIEWERS' COMMENTS:

Reviewer #2 (Remarks to the Author):

I have now read through the authors' responses. They have considered all reviewers' comments very thoroughly. The manuscript is substantially improved, and re-focussed on modern samples, whilst the initial conclusions with fossil enamel are tempered (although still intriguing and promising). The authors make a good point in their response that it is inherently tricky to evaluate $\delta^{15}\text{N}$ enamel against $\delta^{15}\text{N}$ bone-collagen based purely on fossil samples, but that was clearly never their intention, and the modern ground-truthing is so thorough, that this is a minor issue. Indeed, $\delta^{15}\text{N}$ enamel against $\delta^{15}\text{N}$ bone-collagen relationships might in future be a very promising method of identifying diagenesis. This is an exciting line of investigation in its own right. I would particularly love to see a future study with a biplot of $\delta^{15}\text{N}$ offsets vs $\delta^{18}\text{O}$ offsets (enamel vs collagen), potentially also including triple oxygen isotope space. This is now possible, and exciting.

I recommend acceptance, and I congratulate Dr Leichliter and colleagues on an very good piece of science. I am sure that it will be a well-appreciated and well-cited contribution in the field of stable isotope ecology in future years. Well done!

Reviewer #3 (Remarks to the Author):

This is a beautiful contribution with novel results that will be of interest to people in the paleoecology and geochemical community as a whole. Statistical analyses are sound and results are reproducible. The conclusions are well justified and they have incorporated all suggestions provided by reviewers. The result is an elegant solid contribution that reads fluidly. Therefore, I think that this paper is ready for publication without further review. I just found a small typo on line 130 (the word 'from' needs to be removed).